

# Secondary Organic Aerosol from Atmospheric Photooxidation of Indole

Julia Montoya,[1] Jeremy R. Horne,[2] Mallory L. Hinks,[1] Lauren T. Fleming,[1] Veronique Perraud,[1] Peng Lin,[3] Alexander Laskin,[3] Julia Laskin,[4] Donald Dabdub,[2] and Sergey A. Nizkorodov[1]

[1]Department of Chemistry, University of California, Irvine, CA 92697, USA
[2]Department of Mechanical and Aerospace Engineering, University of California, Irvine, CA 92697, USA
[3]Environmental Molecular Sciences Laboratory, Pacific Northwest National Laboratory, Richland, WA 99354, USA
[4]Physical Sciences Division Pacific Northwest National Laboratory, Richland, WA 99354, USA

*Correspondence to*: Sergey A. Nizkorodov (nizkorod@uci.edu)

**Abstract.** Indole is a heterocyclic compound emitted by various plant species under stressed conditions or during flowering events. The formation, optical properties, and chemical composition of secondary organic aerosol (SOA) formed by low-$NO_x$ photooxidation of indole were investigated. The SOA yield ($1.1 \pm 0.3$) was estimated from measuring the particle mass concentration with a scanning mobility particle sizer (SMPS) and correcting it for the wall loss effects. The SOA particles were collected on filters and analysed offline with UV-Vis spectrophotometry to measure the mass absorption coefficient (MAC) of the bulk sample. The samples were visibly brown and had MAC values of ~7 $m^2$/g at λ=300 nm and ~2 $m^2$/g at λ=400 nm, comparable to strongly absorbing brown carbon emitted from biomass burning. The chemical composition of SOA was examined with several mass spectrometry methods. The direct analysis in real time mass spectrometry (DART-MS) and nanospray desorption electrospray high resolution mass spectrometry (nano-DESI-HRMS) were used to provide information about the overall distribution of SOA compounds. High performance liquid chromatography, coupled to photodiode array spectrophotometry and high resolution mass spectrometry (HPLC-PDA-HRMS) was used to identify chromophoric compounds. Indole derivatives, such as tryptanthrin, indirubin, indigo dye, and indoxyl red were found to contribute significantly to the visible absorption spectrum of indole SOA. The potential effect of indole SOA on air quality was explored with the airshed model, which found elevated concentrations of indole SOA during the afternoon hours contributing considerably to the total organic aerosol under selected scenarios. Because of its high MAC values, indole SOA can contribute to decreased visibility and poor air quality.

## 1 Introduction

Atmospheric particulate matter (PM) absorbs and scatters solar radiation and is responsible for diminished visibility in urban areas and for global changes in climate. A primary component of PM is secondary organic aerosol (SOA). While air quality model performance has improved in recent years, disagreements between SOA predictions and measurements remain (Couvidat et al., 2013; Heald et al., 2005; Hodzic et al., 2010; Jiang et al., 2012; Volkamer et al., 2006). This discrepancy may result from incorrect parametrizations of mechanisms for known SOA precursors, as well as from unaccounted



precursors of SOA. Atmospheric researchers have investigated in great detail the SOA generated from oxidation of basic anthropogenic and biogenic volatile organic compounds (VOCs), such as isoprene, monoterpenes, saturated hydrocarbons, and aromatic hydrocarbons. Much less is known about SOA from nitrogen-containing VOCs, even though such VOCs are also common in the atmospheric environment and can potentially provide significant additional pathways for SOA

formation. For example, photooxidation of amines has been shown to serve as a possible SOA source (Price et al., 2014; Silva et al., 2008).

Heterocyclic nitrogen-containing aromatic compounds based on pyrrole, pyridine, imidazole, indole, diazines, purines, etc., have been detected in biomass burning emissions (Laskin et al., 2009). Such compounds can also be emitted by vegetation, for example, indole is produced by wide variety of plants (Cardoza et al., 2003; De Boer et al., 2004; Gols et al., 1999;

McCall et al., 1993; Turlings et al., 1990; Zhuang et al., 2012) and is emitted in response to physical or herbivore-induced stress (Erb et al., 2015; Frey et al., 2004; Misztal et al., 2015; Niinemets et al., 2013; Schmelz et al., 2003; Turlings et al., 2004) and during flowering events (Gentner et al., 2014). Once emitted, indole performs critical roles in plant ecology, for example, in attracting pollinators (Zito et al., 2015). For decades, indole and its derivatives (Figure 1) have been utilized in agriculture, dyes, perfumes, and pharmaceutical applications. One of the better-known derivatives of indole is indigo dye

(also known as indigotin), which is used to dye jeans to their characteristic deep blue colour.

Studies of maize plants revealed that indole acts as an aerial priming agent, released before terpenoids (Erb et al, 2015; Schmelz et al., 2003). For example, Schmelz et al. (2003) examined insect induced volatile emissions in Zea Mays seedlings and demonstrated direct positive relationships between jasmonic acid levels and both sesquiterpene and indole volatile emission. Additionally, they showed that indole can reach near maximal emission levels during nocturnal herbivory and

concluded that indole could function as an early morning signal for parasitoids and predators searching for insect hosts and prey. Niinemets et al. (2013) reviewed several case studies on indole emissions induced by biotic stress and found evidence that there are quantitative relationships between the severity of biotic stress and induced volatile emissions, in addition to the previously demonstrated dose-response relationships for abiotic stresses. Erb et al. (2015) showed that herbivore induced indole emissions in maize plants precede the release of mono-, homo-, and sesquiterpenes, supporting the conclusion that

indole is involved in the airborne priming of terpenoids. Different plant stress mechanisms typically elicit release of the same ubiquitous stress volatile, such as indole, and more stress-specific mono- and sesquiterpene blends (Erb et al., 2015; Genter et al., 2014; Niinemets et al., 2013; Schmelz et al., 2003).

Emissions of indole have also been shown to be well correlated with monoterpene emissions during flowering events (Gentner et al., 2014). Ambient measurements conducted by Gentner et al. (2014) showed that both daytime and night-time

concentrations of indole at their field site in California's San Joaquin Valley were similar to or greater than the dominant monoterpene β-myrcene. The authors stressed the need for future laboratory and modelling studies on the SOA formation potential of indole and other novel compounds identified in their study. A later study by Misztal et al. (2015) used a combination of laboratory experiments, ambient measurements, and emissions modelling to show that plants emit a wide variety of benzenoid compounds (including indole) to the atmosphere at substantial rates, and that current VOC inventories





underestimate biogenic benzenoid emissions. They concluded that emissions of benzenoids from plants are likely to increase in the future due to changes in the global environment and stressed that atmospheric chemistry models should account for this potentially important precursor of SOA.

Despite the importance of indole in the atmospheric environment, only a few studies exist on the mechanism of its photooxidation. Gas-phase oxidation of indole by OH, $O_3$, and $NO_3$ was previously studied by Atkinson et al. (1995). They found that indole reacts with OH and $NO_3$ at collision-limited rates, with rate constants of $1.5 \times 10^{-10}$ cm$^3$ molec$^{-1}$ s$^{-1}$ and $1.3 \times 10^{-10}$ cm$^3$ molec$^{-1}$ s$^{-1}$, respectively. The rate for the reaction of indole with $O_3$ (rate constant $5 \times 10^{-17}$ cm$^3$ molec$^{-1}$ s$^{-1}$) and the rate of direct photolysis were found to be too low to compete with the OH and $NO_3$ reactions. Atkinson et al. (1995) observed 2-formylformanilide as the major primary product of oxidation of indole by both $O_3$ and OH (Fig. 1). Oxidation of indole was also studied by Iddon et al. (1971) in γ-irradiated aqueous solutions, where oxidation by OH was the dominant reaction mechanism. The reaction produced 3-oxoindole, indoxyl red, indirubin, indigo dye, and eventually resulted in a trimer of 3-oxoindole and two indole molecules as the major products.

The formation of SOA from indole has not been previously investigated. The main motivation for investigating the indole SOA is that it may possess unusual optical properties. Many of the indole-derived products are brightly coloured and have distinctive absorption bands in visible spectrum. If these products are formed during atmospheric oxidation of indole and partition into aerosol particles, they can potentially contribute to the pool of organic light-absorbing species. Such organic compounds that absorb radiation strongly in the near-UV and visible spectral ranges are collectively known as "brown carbon" in the atmospheric literature (Andreae and Gelencser, 2006; Laskin et al., 2015).

In this work, we investigate formation of indole SOA in a smog chamber and characterize its molecular composition and optical properties. We incorporate these results in an airshed model with detailed SOA chemistry in order to estimate the effect of indole on the total SOA and on the light-absorbing components of SOA. We show that indole can measurably contribute to SOA loading even in urban environments, where anthropogenic emissions dominate over biogenic ones, such as the South Coast Air Basin of California (SoCAB). Furthermore, we show that indole SOA contains unique strongly-absorbing compounds and can contribute to decreased visibility, especially under plant-stressed conditions or during flowering events.

## 2 Experimental methods

The experiments were performed in a 5 m$^3$ Teflon chamber under low relative humidity (RH < 5%). Hydrogen peroxide was introduced into the chamber by evaporation of a 30 v% solution of $H_2O_2$ in water (Fisher Scientific) into a flow of clean air, to achieve an initial mixing ratio of 2 part per million by volume (ppm). Indole (99% purity, Sigma-Aldrich) was dissolved in methanol (LC-MS grade, 99.9% purity, Honeywell) and was evaporated into the chamber to obtain an initial mixing ratio of 200 parts per billion by volume (ppb), which is equivalent to 960 μg m$^{-3}$. The injector and inlet lines were heated to 70°C to minimize losses on the surfaces. The reported room temperature vapour pressure of indole is 0.012 mmHg (Das et al., 1992), which is equivalent to ~16 ppm. Therefore, the majority of the injected indole should have remained in the gas-





phase. The content of the chamber was mixed with a fan following the injection. The mixing was then stopped and UV-B lamps were turned on to initiate the photooxidation. In some experiments, mixing was not complete by the time the lamps were turned on as evidenced by the measured indole concentrations continuing to increase in the initial photooxidation period. Throughout the experiment, size and number concentration of particles were monitored with a Scanning Mobility

Particle Sizer (SMPS). A Proton-Transfer-Reaction Time-of-Flight Mass Spectrometer (PTR-ToF-MS) monitored the decay of indole, as well as the formation of volatile photooxidation products. When the SOA particles reached a peak concentration in the chamber, the UV irradiation was stopped, and the polydispersed particles were collected on one Teflon filter (47 mm diameter, Millipore FGLP04700) at 20 L min⁻¹ for 3 hours. The amount of the collected SOA material on filters was estimated from SMPS data assuming 100% collection efficiency by the filters and SOA material density of 1.2 g cm⁻³

(Hallquist, et al., 2009).

The SOA yield was calculated from Eq. (1).

$$Yield = \frac{\Delta SOA}{\Delta VOC},\qquad(1)$$

The increase in the mass concentration of particles, $\Delta SOA$, was obtained from SMPS data and corrected for the particle wall loss as described in the supporting information (SI) section. The change in the mass concentration of indole, $\Delta VOC$, was

equated to the initial indole concentration because PTR-ToF-MS data suggested complete removal of indole during the photooxidation.

The filter with the collected sample was cut in half. The first half was used for UV-Vis measurements. The sample was extracted by placing the filter half in a covered petri dish containing 3 mL of methanol (LC-MS grade, 99.9% purity, Honeywell) and shaken vigorously on a shaker for five minutes. The filter colour changed from brown to white suggesting

that most of the light-absorbing compounds were extracted. The methanol SOA extract was then analysed by UV-Vis spectrophotometry in a dual beam spectrometer (Shimadzu UV-2450), with pure methanol used as reference. Wavelength-dependent mass absorption coefficient (MAC) was calculated for indole SOA from the base-10 absorbance, $A_{10}$, of an SOA extract, the path length, $b$ (cm), and the solution mass concentration, $C_{mass}$ (g cm⁻³):

$$MAC(\lambda) = \frac{A_{10}^{solution}(\lambda) \times \ln(10)}{b \times C_{mass}},\qquad(2)$$

The main uncertainty in the calculated MAC values comes from the uncertainty of the mass concentration, which arises from uncertainties in the SMPS measurement of aerosol mass concentration, filter collection efficiency, and solvent extraction efficiency. We estimate that MAC values should be accurate within a factor of two (Romonosky et al., 2015a).

The second half of the filter was used for direct analysis in real time mass spectrometry (DART-MS) measurements. The filter half was extracted in the same way with 3 mL of acetonitrile (LC-MS grade, 99.9% purity, Honeywell). (We elected to

use different solvents for UV-Vis and DART-MS because methanol afforded measurements deeper in the UV region, and acetonitrile gave cleaner background spectra in DART-MS). Aliquots from the acetonitrile SOA extracts were transferred



onto a clean stainless steel mesh, dried in air and manually inserted between the DART ion source and mass spectrometer inlet. The DART-MS consisted of a Xevo TQS quadrupole mass spectrometer (Waters) equipped with a commercial DART ion source (Ion-Sense, DART SVP with Vapur ® Interface). It was operated with the following settings: 350 V grid electron voltage, 3.1 L/min He gas flow, 350°C He gas temperature, and 70°C source temperature. The samples were analysed with

DART-MS in both positive and negative ion modes. Background spectra from pure solvent were also collected and subtracted from the DART mass spectra.

Additional samples were analysed via nanospray desorption electrospray ionization high resolution mass spectrometry (nano-DESI-HRMS) and high performance liquid chromatography, coupled to photodiode array spectrophotometry and high resolution mass spectrometry (HPLC-PDA-HRMS). The former method provides a spectrum of the entire mixture without

prior separation; it is useful for providing an overview of the types of compounds present in SOA. The latter method is suited for advanced detection of individual light-absorbing components in SOA (Lin et al., 2015a,b; Lin et al., 2016). Both methods employ an LTQ-Orbitrap mass spectrometer (Thermo Corp.) with a resolving power of $10^5$ at $m/z$ 400, sufficient for unambiguous characterization of SOA constituents.

For the HPLC-PDA-HRMS measurements, one quarter of the filter was extracted using 350 μL acetonitrile (CH$_3$CN,

gradient grade, ≥99.9% purity) and the change in filter colour from brown to white suggested that most light-absorbing compounds were extracted into the solution. Separation of the SOA extract was achieved with a Scherzo SM-C18 column (Imtakt USA). The gradient elution protocol included a 3 min hold at 10% of CH$_3$CN, a 45 min linear gradient to 90% CH$_3$CN, a 16 min hold at this level, a 1 min return to 10% CH$_3$CN, and another hold until the total scan time of 90 min. The column was maintained at 25 °C and the sample injection volume was 8 μL. The UV−Vis spectrum was measured using

PDA detector over the wavelength range of 250 to 700 nm. The ESI settings were: positive ionization mode, + 4.5 kV spray potential, 35 units of sheath gas flow, 10 units of auxiliary gas flow, and 8 units of sweep gas flow.

The HRMS data analysis was performed by methods summarized by Romonosky at al. (2015b). Briefly, the mass spectra were clustered together, the $m/z$ axis was calibrated internally with respect to expected products of photooxidation, and the peaks were assigned to formulas C$_c$H$_h$O$_o$N$_n$Na$_{0-1}^+$ or C$_c$H$_h$O$_o$N$_n^-$ constrained by valence rules and elemental ratios ($c,h,o,n$

refer to the number of corresponding atoms in the ion). These were then converted to formulas of the corresponding neutral species, obtained by removing Na or H from the observed positive ion formulas, or adding H to the negative ion formulas. The HPLC-PDA-HRMS was done as described in Lin et al. (2015b, 2016).

## 3 Modelling methods

Air quality simulations were performed to complement laboratory experiments and to assess the formation of indole SOA in

a coastal urban area. The University of California, Irvine – California Institute of Technology (UCI-CIT) regional airshed model with a state-of-the-art chemical mechanism and aerosol modules was used in this study. The model domain utilized 4970 computational cells (five vertical layers with 994 cells per layer) with a 5 km x 5 km horizontal grid size and





encompassed the South Coast Air Basin of California (SoCAB), including the Pacific Ocean on the west side, heavily populated urban areas around Los Angeles, and locations with a high density of plant life such as the Angeles National Forest on the east side. The model included spatially and temporally resolved emissions and typical meteorological conditions for this region. The emissions inventory used in this study was based on the 2012 Air Quality Management Plan

(AQMP) provided by the South Coast Air Quality Management District (SCAQMD, 2013). Boundary and initial conditions were based on historical values. Simulations were performed for a 3-day summer episode. Two days of model spin-up time were used to reduce the influence of initial conditions and allow sufficient time for newly added emissions to drive changes in air quality. Results shown below are for the third day of the simulations.

The UCI-CIT model utilizes an expanded version of the Caltech atmospheric chemical mechanism (CACM; Dawson et al.,

2016; Griffin et al., 2002a,b; Griffin et al., 2005) and has been used in numerous other studies to simulate air quality in the SoCAB (Carreras-Sospedra et al., 2006; Carreras-Sospedra et al., 2010; Chang et al., 2010; Nguyen and Dabdub, 2002). The CACM includes a comprehensive treatment of SOA known as the Model to Predict the Multiphase Partitioning of Organics (MPMPO) (Griffin et al., 2003; Griffin et al., 2005). MPMPO is a fully coupled aqueous/organic equilibrium-partitioning-based model and is used to calculate gas-particle conversion of secondary organic species. The SIMPOL.1 group-

contribution method of Pankow and Asher (2008) is used to calculate vapour pressures of SOA species for use in MPMPO. Activity in both the aqueous and organic phases is determined using the UNIFAC model of Hansen et al. (1991). Henry's Law constants are calculated according to the to the group contribution method of Suzuki et al. (1992). Several studies have used the UCI-CIT model to investigate secondary organic aerosol formation, dynamics, reactivity, and partitioning phase preference in the SoCAB (Carreras-Sospedra et al., 2005; Chang et al., 2010; Cohan et al., 2013; Dawson et al., 2016;

Griffin et al., 2002b; Vutukuru et al., 2006). For a more detailed description of recent model developments incorporated into the UCI-CIT model and its SOA modules, the reader is referred to Dawson et al. (2016).

For the present study, the chemical mechanism was modified from the base case version to include species and processes shown in Figure 2. Two new gas-phase species were added: indole and its representative oxidation product. Because of the high mass yield of indole SOA, any reasonable indole oxidation product with a low vapour pressure would be suitable. We

elected to use indigo dye ($C_{16}H_{10}N_2O_2$) because it is a very common derivative of indole and because its formula was reasonably close to the average formula of SOA compounds determined from nano-DESI ($C_{15}H_{11}O_3N_2$). One new gas-phase reaction was added, which forms gas-phase indigo dye via oxidation of gas-phase indole by hydroxyl radical. Lastly, indigo dye was also added to the model as a new SOA species. Gas-phase indigo dye was assumed to partition into the aerosol phase based on its calculated vapour pressure and Henry's Law constant. After the modifications described here, the model

contained a total of 202 gas-phase species, 607 gas-phase reactions, and 18 SOA species. Each SOA species was sorted into eight distinct size bins based on particle diameter, up to a maximum of 10 µm. The activity coefficient of indigo dye was assumed to be equal to one.

Because gas-phase indole was not included in the base case emissions inventory, its emission rate was estimated based on available literature data. As discussed in the introduction section, emissions of indole have been shown to be well correlated


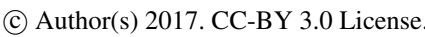


to emissions of monoterpenes in a variety of plant species (Erb et al., 2015; Gentner et al., 2014; Niinemets et al., 2013). However, most existing data were obtained from controlled laboratory experiments and emissions of indole at the regional scale are not well constrained. In this work, emissions of gas-phase indole were added to the base case emissions inventory by using a ratio to an existing gas-phase species in the emissions inventory, BIOL. BIOL is representative of lumped

biogenic monoterpenes and contains spatially and temporally resolved emissions in the base case inventory. Therefore, the spatiotemporal distribution of indole emissions follows that of BIOL, with the magnitude of the emissions set to a given percentage of BIOL emissions. No direct emissions of gas-phase indigo dye were added to the model. Because of the uncertainty and episodic nature of gas-phase indole emissions, simulations were performed with a range of possible emission factors to determine the sensitivity of indole SOA formation to gas-phase indole emissions.

Four scenarios were considered for model calculations. The first scenario had zero emissions of gas-phase indole. This scenario will be referred to as the "base case" and serve as the reference scenario to which the other scenarios are compared to determine changes in air quality. The remaining scenarios had emissions of gas-phase indole set to 5%, 10%, and 25% of BIOL emissions, referred herein as "low", "medium", and "high" emissions, respectively. When averaged over the entire simulation domain, the corresponding average emission factors for indole were 0.25, 0.51, and 1.27 $\mu$g m$^{-2}$ h$^{-1}$, respectively.

Similar emission factor of 0.6 $\mu$g m$^{-2}$ h$^{-1}$ for indole was used in previous study of Misztal et al. (2015) where indole emissions under average stress conditions were incorporated in the MEGAN 2.1 biogenic VOC emissions model to estimate total global emissions. Therefore, the medium emission scenario considered in this study should be representative of the emissions of indole under average stress conditions, while the high emissions scenario is more likely to represent episodic emission events such as those during springtime flowering or herbivore infestation.

## 4 Results and discussion

### 4.1 Properties of indole SOA

Figure 3 illustrates the time dependence of mass concentrations of indole and particulate matter in a typical chamber experiment. According to PTR-ToF-MS measurements, indole decayed with a half-life of approximately 60 min, which translates into an average OH concentration in the chamber of $1.4\times10^{6}$ molec cm$^{-3}$, similar to ambient levels (Fig. S2.1). The

25 PTR-ToF mass spectrum of indole before photooxidation (Fig. S2.2) was dominated by the protonated indole at *m/z* 118.067 (the *m/z* values cited in the text correspond to the measured *m/z* values; the corresponding exact *m/z* values are listed in Table S2). After photooxidation, a few other prominent peaks appeared. Figures S2.3, S2.4, and S2.5 show the time-dependence profiles of several peaks of interest detected by PTR-ToF-MS during the photooxidation of indole, and Table S2 contains their proposed assignments. Peaks at *m/z* 120.072, 131.062, and 132.050 (Figure S2.4) appeared simultaneously with indole

injection, suggesting that the indole sample contained small amounts (<2%) of indoline, diazanaphthalene, and 3-oxyindole impurities, respectively, which may have contributed to SOA formation. From the ions that first appeared and then were consumed during photooxidation (Figure S2.2), the one at *m/z* 122.061 had the largest peak abundance. It corresponds to protonated 2-formylformanilide [M+H]$^{+}$ ion (Figure 1), a major gas-phase product of indole oxidation by OH (Atkinson et



al., 1995). Another significant product was detected at *m/z* 148.041 and was tentatively assigned to the [M+H]$^+$ ion from isatin (Figure 1). The unusual time dependence for this peak shown in Figure S2.3 was reproducible, and implies complex mechanism for the production and removal of isatin. Isatin also was observed as an abundant peak in both DART(+) and nano-DESI(+) mass spectra, suggesting the it can be partitioned between the gas and particle phases. Smaller peaks produced
and then consumed in photooxidation included indoxyl, benzonitrile, and phenylamine. A few peaks at smaller *m/z* grew during the photooxidation (Figure S2.5) including cyanic acid, acetaldehyde, acetone, and acetic acid.

The particles had a geometric mean diameter of approximately 0.3 µm when the filter collection started. The terminal wall-loss corrected mass concentration of particles (Figure 3) was almost the same as the initial concentration of indole suggesting that the SOA yield, defined by Eq. (1), was high. For five experiments repeated under the same conditions on separate days,
the SOA yields calculated from Eq. (1) were 1.04, 0.94, 0.74, 1.52, and 1.25 with an average of 1.1 ± 0.3. We normally get much more reproducible yields for more volatile precursors, such as monoterpenes, so we attribute the large scatter in the yield of indole SOA to losses of indole in the injector. Therefore, the reported average value of 1.1 likely represents a lower limit for the actual yield. The high yield is comparable to that for SOA formed from another bicyclic aromatic compound, naphthalene, which has a reported yield of 0.73 under low-NO$_x$ conditions (Chan et al., 2009).

Figure 4 shows the MAC values measured for an extract of indole SOA in methanol. MAC reached values of ~7 m$^2$/g at λ = 300 nm. At λ = 400-600 nm, the MAC values ranged from 2 to 0.1 m$^2$/g, approximately 20-1% of the MAC values reported for black carbon (Knox et al., 2009; Martins et al., 1998). The wavelength dependence of MAC deviates from the power law commonly observed for brown carbon (e. g. see reviews of Laskin et al., 2015; Moise et al., 2015) and has a reproducible broad band at ~350 nm, possibly due to the well-known derivatives of indole, indirubin, indigo dye, and indoxyl red, which
have characteristic absorption bands at this wavelength (see below). For the wavelength range of 300-600 nm, the absorption Angstrom exponent was ~6, comparable to the value of ~5 reported for brown carbon from biomass burning  (Kirchstetter et al., 2012 #3725).

We used two offline MS methods (DART and nano-DESI) and both negative and positive ion modes to characterize the SOA composition in order to detect a broader range of compounds than possible with a single method. Figure 5 shows the
DART and nano-DESI mass spectra of indole SOA in both positive and negative modes. The high resolving power of nano-DESI-HRMS afforded unambiguous formula assignments for all peaks up to *m/z* 500, and the molecular weights (MWs) of the neutral compounds could be determined from the corresponding ion formulas. About half of the ions observed in nano-DESI (+) mass spectra were [M+Na]$^+$ species, and the remaining compounds were protonated molecules, [M+H]$^+$. The DART mass spectra were acquired on a triple quadrupole mass spectrometer much lower resolving power. As a result, only
the nominal *m/z* values for the observed peaks could be determined. It was assumed that the dominant mechanism of ionization was protonation ([M+H]$^+$ ions formed; nominal MW = nominal *m/z* – 1) in the positive ion mode and deprotonation ([M-H]$^-$ ions formed; nominal MW = nominal *m/z* + 1) in the negative ion mode (Nah et al., 2013). For ease of comparison, all the mass spectra were plotted as function of the exact mass of the corresponding neutral compounds.



For a given ion mode, the DART and nano-DESI mass spectra were qualitatively similar, although nano-DESI appeared to favour larger, more oxidized compounds compared to DART. Both DART and nano-DESI mass spectra showed a clear separation into distinct clusters of peaks corresponding to monomer, dimer, trimer, and tetramer oxidation products. For a given ion mode, the major monomer peaks were the same in DART and nano-DESI strongly suggesting that they correspond to more abundant indole oxidation products (as opposed to minor SOA compounds that happened to have unusually high ionization efficiencies). There is also good correspondence between the major dimer peaks recorded in DART and nano-DESI. In both DART and nano-DESI mass spectra, the peak abundances in the negative ion mode spectra are shifted towards higher molecular weights compared to the positive ion mode mass spectra. The preferential negative ion formation from more oxidized compounds was previously observed in ESI mass spectra of limonene SOA (Walser et al., 2008). Although we cannot assign formulas to the DART-MS peaks, it is evident from Figure 5 that this ionization method also favours larger, and presumably more oxidized, compounds in the negative ion mode. For example, carboxylic acids are more readily observed in the negative ion mode DART mass spectra (Nah et al., 2013).

Table 1 lists the most abundant peaks observed in the monomer and dimer ranges of nano-DESI-HRMS and DART-MS data, as well as additional smaller peaks for the specific compounds discussed in this paper. Isatin ($C_8H_5O_2N$, MW = 147 Da) was the single dominant peak in the monomer range observed in both nano-DESI(+) and DART(+); it was also detected in the negative ion mode mass spectra. Isatoic anhydride ($C_8H_5O_2N$; MW = 163 Da) was the second most abundant monomeric peak in all four mass spectra. Other abundant monomeric products included 3-oxyindole ($C_8H_5ON$; MW = 131 Da) and 2-formylformanilide ($C_7H_7ON$; MW=121 Da). Of the compounds shown in Figure 1, tryptanthrin ($C_{15}H_{10}O_2N_2$; MW = 250 Da), indirubin ($C_{16}H_{10}O_2N_2$; MW= 262 Da), indigo dye ($C_{16}H_{10}O_2N_2$; MW = 262), and dihydro indigo dye ($C_{16}H_{12}O_2N_2$; MW = 264 Da) were the most abundant dimer peaks. Meanwhile, indoxyl red ($C_{16}H_{10}ON_2$; MW = 246 Da) was detected with lower but appreciable abundances in nano-DESI(-) and in both DART mass spectra. The prominent dimer compounds listed in Table 1 contained additional oxygen atoms compared to indoxyl red, indirubin, indigo dye, and dihydro indigo dye, and could be formed by further oxidation of these compounds.

Figure 6 shows the distribution of the number of C atoms in the indole SOA compounds, as detected by nano-DESI-HRMS (for each group of compounds with the same number of C atoms, the abundances in the positive and negative ion mode mass spectra were added together). The majority of the observed compounds contained 8, 16, or 24 C atoms, corresponding to the monomer, dimer, and trimer derivatives of indole. Peaks with 7 and 15 carbon atoms were also prominent, suggesting an important role of the primary $C_7$ oxidation product 2-formylformanilide in the formation of low volatility species. Minor peaks containing other C-numbers were also present suggesting further fragmentation of the primary oxidation products. The average formula for all observed SOA compounds was $C_{15}H_{11}O_3N_2$.

Figure S3 shows the distribution of the N/C ratios in indole SOA compounds. A large number of the compounds had the same N/C ratio as indole (N/C = 1/8) indicating the oxidation and oligomerization reactions conserved both N and C atoms in many of the products. However, some products had a slightly larger ratio consistent with a loss of C atoms (e.g., N/C = 1/7 and 2/15), whereas some products gained extra C atoms. One product with a relatively large abundance, $C_{12}H_{14}O_4$, had





no N atoms left in it. In addition, there were several $C_{8-9}H_hO_oN_2$ products, which gained an additional N atom. The mechanism of photooxidation is clearly complex involving a large number of secondary reactions. The full mechanism of indole photooxidation cannot be obtained from this data set. In the discussion that follows, the focus will be on the mechanism of formation of light-absorbing products.

Figure 7 shows the HPLC-PDA chromatogram of an indole SOA sample demonstrating its components with strong light-absorbing properties near-UV and visible spectral ranges (above 300 nm). In order to identify specific chromophores from the HPLC-PDA-HRMS data, the methods described by Lin et al. (2015b, 2016) were followed. High-resolution mass spectra were examined to identify *m/z* values that appear at the retention times associated with the peaks in the LC chromatograms. The PDA absorption spectra associated with these retention times were then compared with possible candidates constrained

by their molecular formula determined from the mass spectra.

Figure 8 shows a comparison of the absorption spectra for the key peaks in the HPLC-PDA chromatogram with absorption spectra of selected compounds reported in the literature. The match is excellent in terms of the absorption peak maxima: 280, 310, 334, 392 nm for tryptanthrin; 240, 283, 335, 610 nm for indigo dye; 242, 290, 365, 540 nm for indirubin; and 217, 273, 350, 520 nm for indoxyl red. The shapes of the spectra do not match perfectly because the chromophores are not fully

separated by the HPLC column (Figure 7) and may co-elute with additional minor compounds. Likely, more than one chromophore contributed to the absorbance at any given retention time. However, the power of the method is clear, as illustrated, for example, by the distinction of the structural isomers indigo dye and indirubin (Figure 7).

The precursors to indoxyl red and indigo dye, dihydro indoxyl red and dihydro indigo dye, respectively, were also identified by this analysis, and were observed in nano-DESI mass spectra. This observation supports the mechanism of aqueous-phase

indole oxidation proposed by Iddon et al. (1971), in which indole first oxidizes to 3-oxidole, then to dyhydro indoxyl red or dyhydro indigo dye, then finally to indoxyl red and indigo dye (Fig. 9a). Although the mechanism by Iddon et al. (1971) was developed for the aqueous oxidation of indole, it appears consistent with the photooxidation of indole in gaseous phase.

Several products were assigned based on previous observations by Novotna et al. (2003). They proposed the mechanism shown in Figure 9b to explain the production of tryptanthrin and anthranilic acid from ambient indigo dye oxidation. In this

mechanism, hydroxyl radicals attack the carbonyl carbon atoms of isatin ultimately opening the 5-membered N-heterocyclic ring to yield anthranilic acid. Although anthranilic acid does not show up in Figure 7 because it is not a chromophoric species, it was detected by nano-DESI-HRMS. As shown in Figure 9b, anthranilic acid can react with another molecule of isatin to produce tryptanthrin. This mechanism is particularly relevant to indole SOA, because isatin can be produced not only from the oxidation of indigo dye but also directly from indole, through the intermediacy of 3-oxindole (Fig. 9a).

Moreover, Novotna and colleagues suggested that isatoic anhydride should be formed from indigo dye oxidation. A compound with this formula had large abundance in both HPLC-PDA-HRMS (Figure 9) and nano-DESI-HRMS and DART-MS (Table 1). Combined with the evidence that tryptanthrin is a major secondary chromophore, this could be a significant pathway to brown carbon formation in the oxidation of indole.



## 4.2 Potential effects of indole SOA

The spatiotemporal distribution of indole SOA is likely controlled by a combination of: (i) the spatiotemporal distribution of gas-phase indole and its emissions sources, (ii) the availability of hydroxyl radical for gas-phase oxidation chemistry and (iii) meteorological conditions in the region, including temperature, humidity, and wind direction. Once emitted, indole reacts

with hydroxyl radical to form gas-phase indigo dye. Gas phase indigo dye can then partition into the aerosol phase to form indole SOA. The presence of a sea breeze in the SoCAB results in a prevailing wind direction of north-northeast, transporting pollutants inland during the daytime hours. As a result, peak concentrations of indole SOA should be located further inland than peak concentrations of gas-phase indole and occur in areas that are already burdened with poor air quality.

Figure S4 shows the spatial distribution of 24-hour average gas-phase indole concentrations in the SoCAB for the four modelled scenarios considered in this study. The amount of indole SOA formed in the model, and thus the impact of indole on the total predicted SOA concentrations, depends strongly on the emissions of gas-phase indole. In the high emissions scenario, hourly gas-phase indole concentrations peaked at 0.3 ppb, with the highest concentrations occurring in the early morning hours before sunrise. For comparison, during a field measurement campaign in the San Joaquin Valley of

California, Gentner et al. (2014) reported gas-phase indole concentrations of about 1-3 ppb in ambient air during a springtime flowering event. Measured concentrations of indole were slightly higher during the late night and early morning hours than during the daytime, consistent with the model results obtained in this study. Gentner et al. (2014) also showed that flowering was a major biogenic emissions event, causing emissions of many compounds to increase by several factors to over an order of magnitude. Therefore, episodic emissions of indole in rural areas are likely to be significantly greater than

the emissions used in this study. Based on the high SOA yield from gas-phase indole found in this study we propose that biogenic emissions events such as springtime flowering may degrade local air quality.

Figure 10a shows 24-hour average SOA concentrations in the base case model simulation, and Figures 10b, 10c, 10d show the additional SOA resulting from indole in the three emissions scenarios. The highest SOA concentrations occurred directly east of Riverside where a combination of biogenic and anthropogenic precursors accumulated during days one and two and

into day three. The 24-hour average indole SOA concentrations peaked at about 0.13 µg/m³ in the high emissions scenario (Fig. 10d). The highest concentrations of indole SOA occurred north of Los Angeles and Riverside. To put this number in perspective, aerosol with mass concentration of 0.1 µg/m³ and MAC of 1 m²/g will have an absorption coefficient of 0.1 Mm⁻¹ (we neglect the particle size effects in this estimation). Thompson et al. (2012) reported an absorption coefficient of 4 Mm⁻¹ at 532 nm during the 2010 CalNex campaign in Pasadena, California, with the absorption being dominated by BC. The

average absorption coefficients reported for "average urban USA" and "average remote USA" by Horvath et al. (1993) were 22 Mm⁻¹ and 0.7 Mm⁻¹, respectively. While the absorption by indole SOA is unlikely to compete with that by black carbon in urban areas, it may contribute to the aerosol absorption in more remote areas, where the black carbon concentrations are smaller.



SOA concentrations averaged over the entire domain are shown in Figure 11 for all four modelled scenarios. The averaged SOA concentrations were computed by averaging the concentration of total SOA in all computations cells in the domain. Therefore, changes in the averaged SOA concentrations are representative of the overall impact on total SOA concentrations for the entire basin. In the high emissions scenario, the averaged SOA concentrations increased by about 4-13%, indicating

that indole SOA can contribute significantly to total organic aerosol concentrations. While base case SOA concentrations peaked during the early morning and late night hours when metrological conditions were favourable, the largest *changes* in SOA concentrations occurred during the late morning and afternoon hours. The formation of gas-phase indigo dye and indole SOA depends on the availability of the hydroxyl radical, which reaches peak concentrations during daylight hours when photochemistry is active. Therefore, increased production of hydroxyl radical during the daytime accelerates the

oxidation of gas-phase indole, ultimately resulting in increased formation of indole SOA. Increases in total SOA are due mostly to the formation of indole SOA, with only small changes in the concentration of other SOA species.

The amount of indole SOA formed in each scenario was found to be directly proportional to the emissions of gas-phase indole. In the low emissions scenario, gas-phase indole and indole SOA concentrations were about factor of five lower than those seen in the high emissions scenario, with 24-hour average indole SOA concentrations peaking at about 0.025 µg/m$^3$.

Similarly, relative increases in the averaged SOA concentrations ranged from 1-3% in the low emissions scenario. In the medium emissions scenario, 24-hour average indole SOA concentrations reached about 0.05 µg/m$^3$, causing total SOA concentrations to increase by 2-6%. In all three emissions scenarios, the spatial distribution of indole SOA remained essentially the same, with peak concentrations occurring in the northeast portion of the basin, an area already burdened with poor air quality.

**5 Conclusions**

This work demonstrates that indole is an effective precursor to SOA. At the concentrations used in this chamber study, the majority of indole oxidation products ended up in the particle phase, with an effective SOA yield of ~1.1 ± 0.3. The resulting SOA was found to be highly light-absorbing, with MAC values (~ 1 m$^2$/g) approaching those for strongly-absorbing brown carbon from biomass burning. The high MAC values were due to several well-known chromophoric products of indole

oxidation, including tryptanthrin, indirubin, indigo dye, and indoxyl red, which were identified by their molecular formulas and characteristic peaks in their absorption spectra. These observations suggest that N-heterocyclic compounds may be important contributors to secondary BrC.

Contribution of indole to SOA formation can potentially result in reduced visibility, particularly in regions where plants are exposed to biotic and abiotic stresses. When combining the experimental MAC values with peak SOA concentrations

predicted in the model, the estimated absorption coefficient is 0.1 Mm$^{-1}$ due to indole SOA. This is smaller than the values typically reported for SoCAB but comparable to values reported in more remote areas. Thus, despite its large MAC, indole SOA is not likely to contribute to particle absorption in urban areas, where the aerosol absorption is dominated by



anthropogenic black carbon. However, the situation could be different in remote areas, where black carbon does not contribute to aerosol absorption, and indole emissions are higher.

The UCI-CIT regional airshed model demonstrated significant potential for indole SOA formation. While the mass loading of indole SOA in the SoCAB was relatively low in all scenarios, it represents a previously unconsidered source of SOA in air quality models, which have improved in recent years, but still tend to disagree with measured SOA concentrations (Couvidat et al., 2013; Heald et al., 2005; Hodzic et al., 2010; Jiang et al., 2012; Volkamer et al., 2006). Indole SOA can interact with other aerosol phase species, causing indirect changes in the concentration of total SOA. Such interactions were not considered in this study because an activity coefficient of unity was used for indole SOA in the model simulations. Rural or agricultural regions with significant biomass burning or a high density of plant life likely have much higher emissions of gas-phase indole than the SoCAB. For example, field measurement studies (Gentner et al., 2014) reported ambient indole concentrations up to an order of magnitude greater than the peak modelled concentrations employed in this study, indicating a significant potential for indole SOA formation in rural areas. Furthermore, future climate change is likely to increase gas-phase indole emissions through environmental and physical stress factors such as drought, elevated temperatures, increased $CO_2$ and $O_3$ concentrations, and enhanced herbivore feeding (Yuan et al., 2009). Therefore, indole represents a potentially important source of biogenic SOA that is currently unaccounted for in regional and global models.

**Competing interests**

The authors declare that they have no conflict of interest.

**Acknowledgements**

This research was enabled by funding from the United States Environmental Protection Agency under grant EPA 83588101. Julia Montoya acknowledges support from the California LSAMP Bridge to the Doctorate Program at the University of California, Irvine, which is funded by grant NSF-1500284. The DART-MS and PTR-ToF-MS instruments used in this work were purchased with grants NSF CHE-1337080 and NSF MRI-0923323, respectively. The HRMS measurements were performed at the W.R. Wiley Environmental Molecular Sciences Laboratory (EMSL) – a national scientific user facility located at PNNL, and sponsored by the Office of Biological and Environmental Research of the U.S. DOE. PNNL is operated for U.S. DOE by Battelle Memorial Institute under Contract No. DE-AC06-76RL0 1830.

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



**Figure 1:** Chemical structures, common names, molecular formulas, and nominal molecular weights (MW) for indole and its oxidized derivatives discussed in this work.





**Gas-phase**

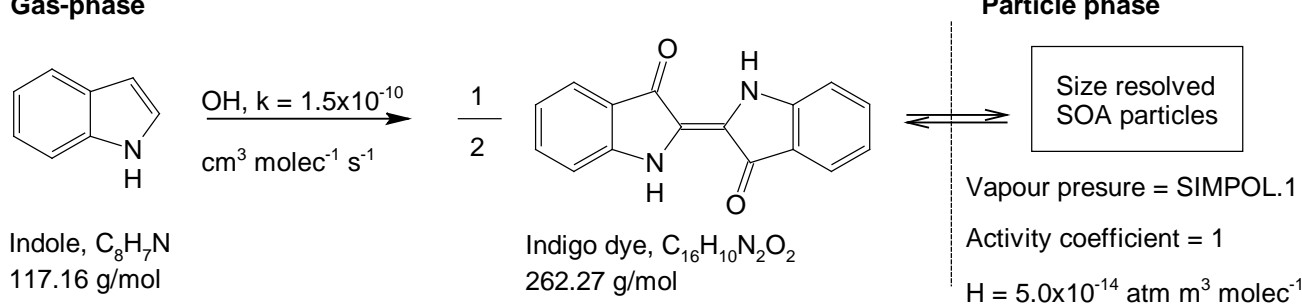

Indole, $C_8H_7N$
117.16 g/mol

Indigo dye, $C_{16}H_{10}N_2O_2$
262.27 g/mol

**Particle phase**

Size resolved
SOA particles

Vapour presure = SIMPOL.1

Activity coefficient = 1

H = $5.0 \times 10^{-14}$ atm m$^3$ molec$^{-1}$

OH, k = $1.5 \times 10^{-10}$ cm$^3$ molec$^{-1}$ s$^{-1}$

$\frac{1}{2}$

**Figure 2:** Summary of modifications made to the UCI-CIT model chemical mechanism and aerosol modules.



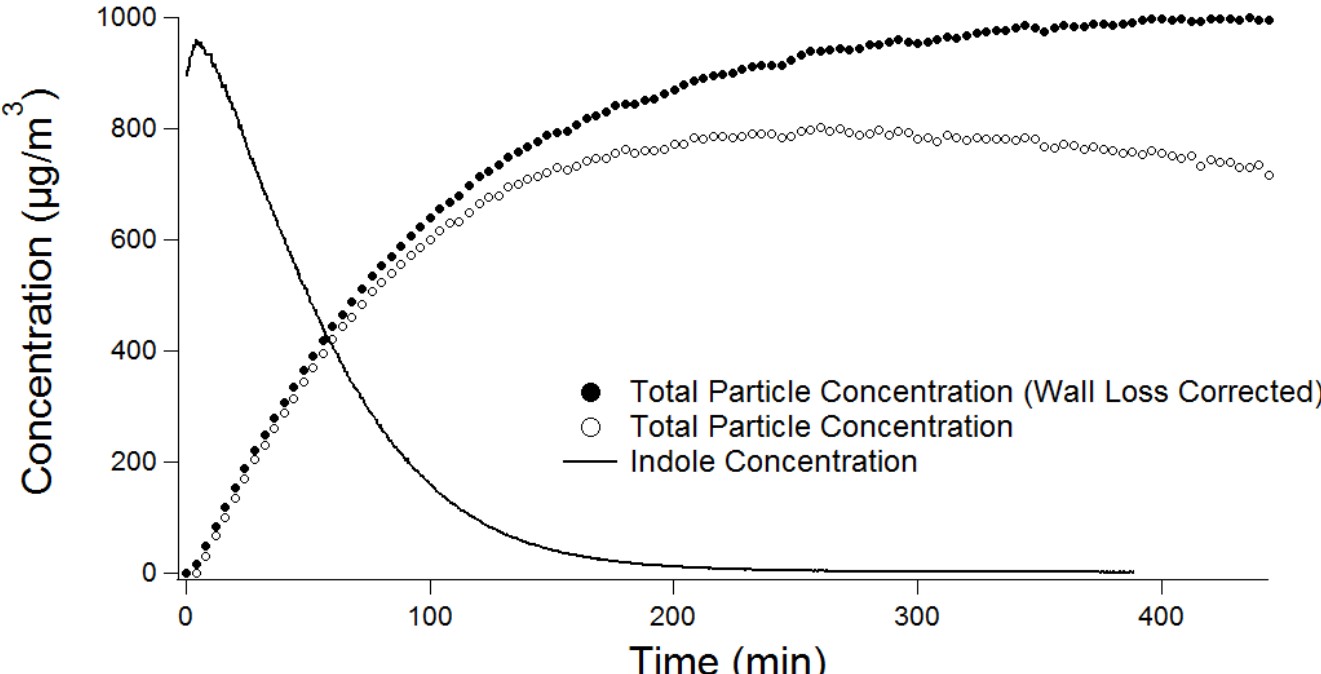

**Figure 3:** The mass concentration of indole (solid trace), the mass concentration of particles (open circles), and the wall loss corrected mass concentration of particles (solid circles) over time. Indole was not yet fully mixed in the chamber by the time photooxidation started at t=0 resulting in an apparent initial rise in the measured indole concentration.




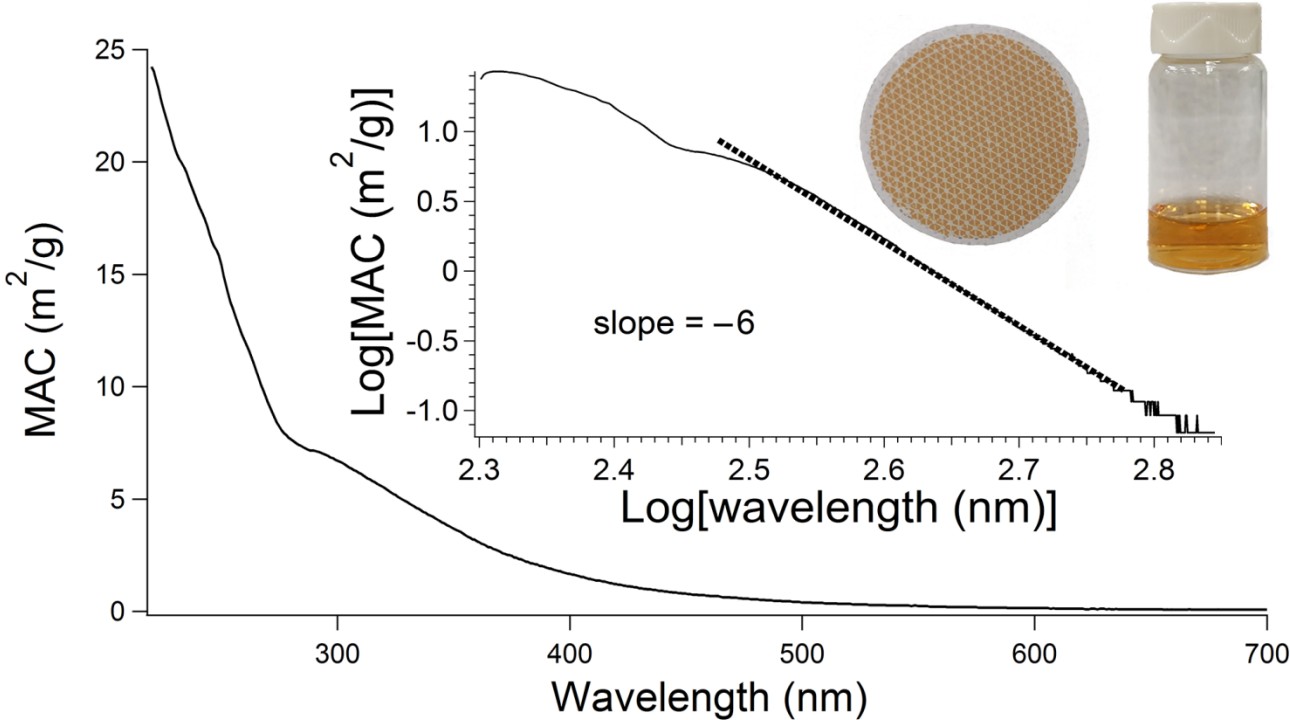

**Figure 4:** Wavelength-dependent mass absorption coefficient (MAC) of indole SOA. The inset shows the log-log version of the same data used to determine the absorption Angstrom exponent (fitted from 300 to 600 nm) as well as photographs of the indole SOA collected on a filter and extracted in methanol.





**Figure 5:** nano-DESI and DART mass spectra of indole SOA plotted as a function of the molecular weights of the neutral compounds. The nano-DESI mass spectra contained only peak assignable to specific formulas, while DART mass spectra peaks contain all observed peaks.

*Note to editors: use full page width for this figure*





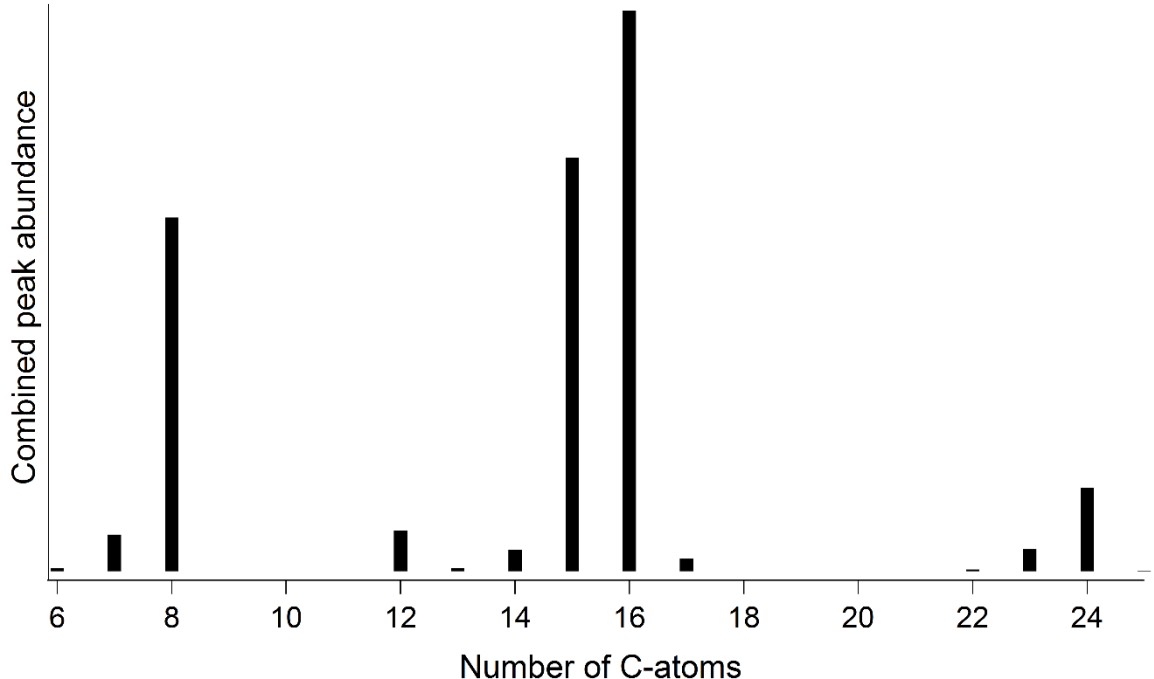

**Figure 6:** Distribution of the number of C atoms in the indole SOA compounds detected in both positive and negative ion model nano-DESI.



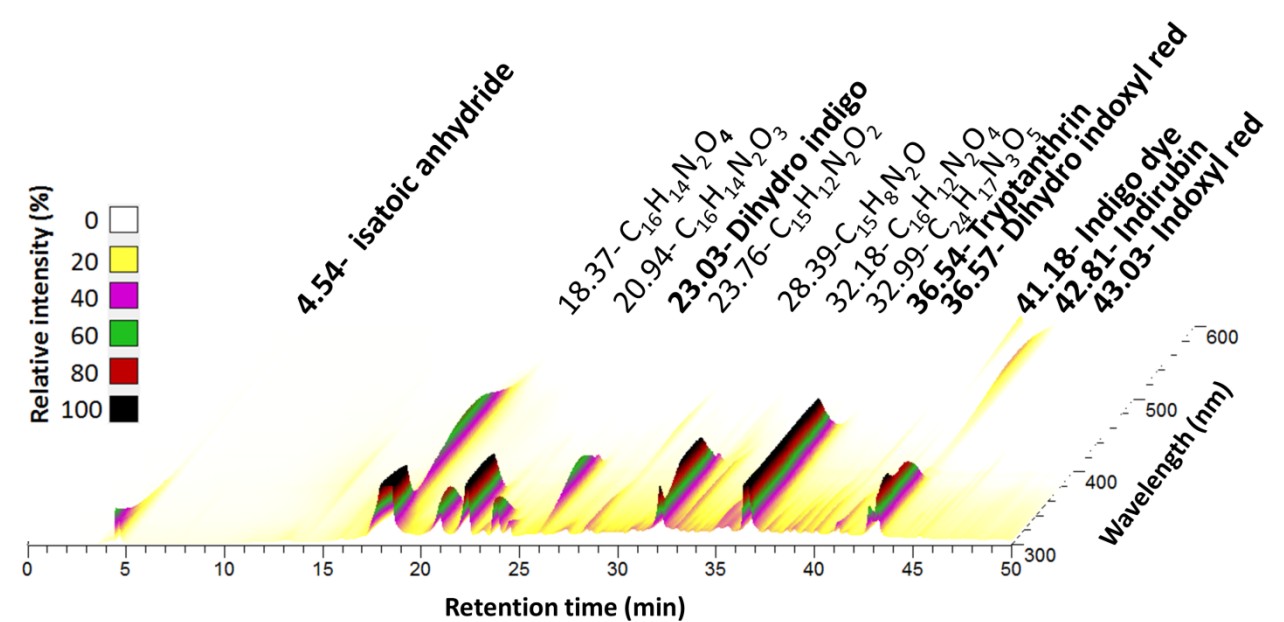

**Figure 7**: HPLC-PDA chromatogram of indole SOA. The absorbance is plotted as a function of both retention time and wavelength. Peaks are labelled by their PDA retention time followed by their proposed assignment. Bold-faced assignments are specific isomers that are discussed further in the text.

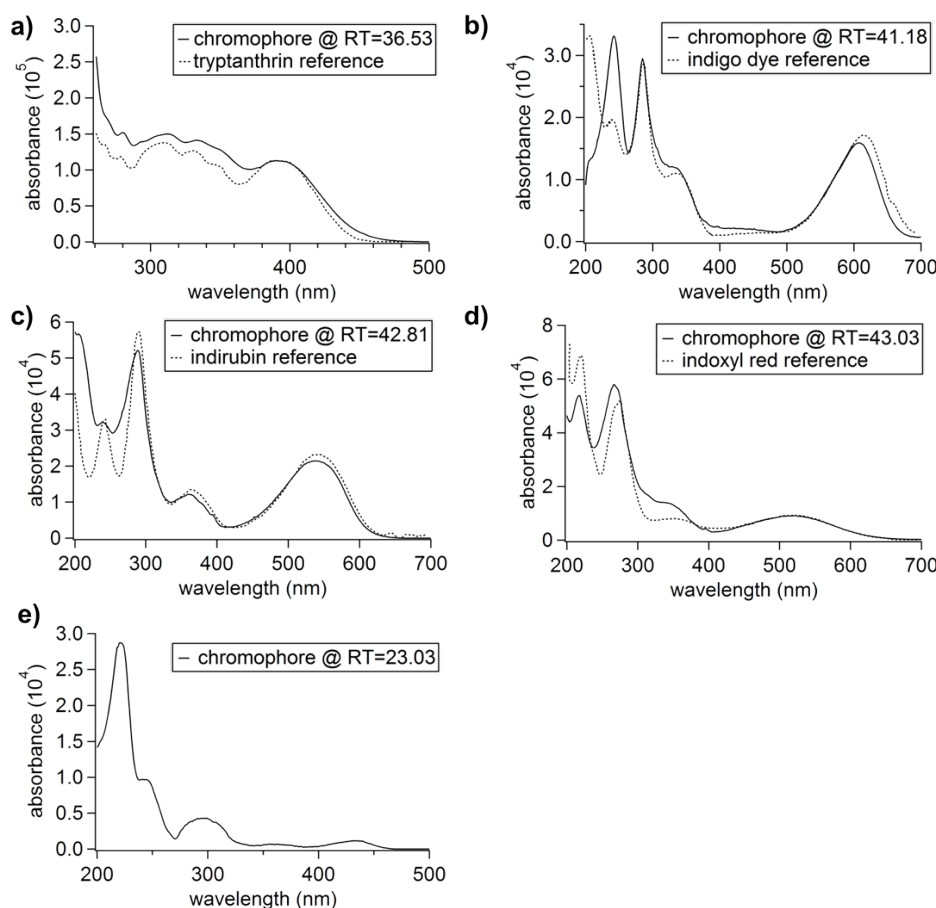

**Figure 8:** Comparison between measured PDA absorption spectra at selected retention time (RTs) and reference spectra of proposed chromophores in the literature (reference spectrum is not available for dihydro ingigo dye, panel e).





**Figure 9:** Processes leading to the formation of observed chromophores in the photooxidation of indole. (a) Processes leading to indigo dye and indoxyl red based on Iddon et al. (1971). (b) Processes leading to tryptanthrin based on Novotna et al. (2003). "Ox" denotes an oxidation step.




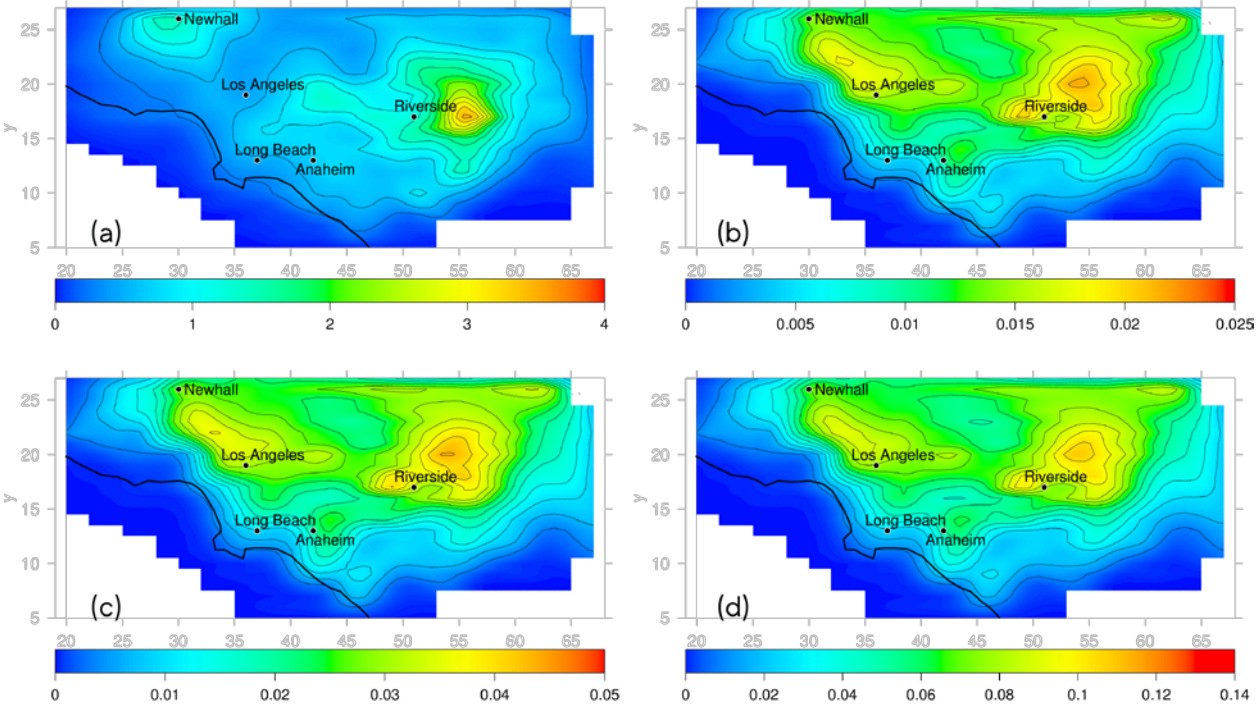

**Figure 10:** 24-hour average concentrations (μg/m$^3$) of (a) total SOA in the base case, (b) indole SOA in the low emissions scenario, (c) indole SOA in the medium emissions scenario, and (d) indole SOA in the high emission scenario.



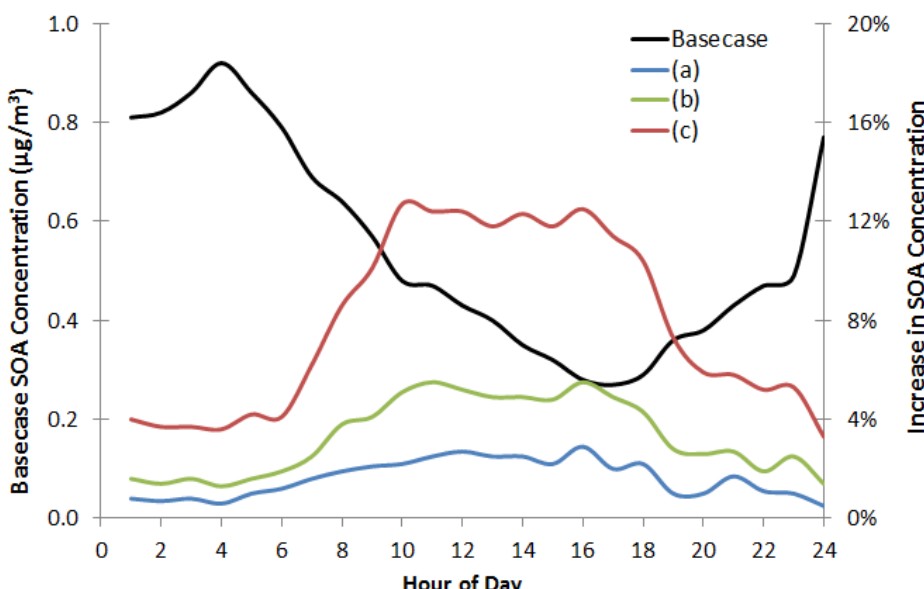

**Figure 11:** Domain wide average SOA concentrations in the base case (black line, left axis) and the relative increase in domain wide average SOA concentrations (right axis) in the (a) low emissions scenario, (b) medium emissions scenario, and (c) high emissions scenario.



**Table 1:** Monomer and dimer peaks with the largest peak abundance observed in DART-MS and nano-DESI-HRMS data. Selected peaks corresponding to the compounds shown in Fig. 1 are also included. Proposed assignments are based on the formulas from nano-DESI-HRMS. Peak abundances are normalized with respect to the most abundant peak in each spectrum.

| | Nominal Mass | Formula | Ionization by $H^+$ or $Na^+$ in nano-DESI (+) | Peak Abundance nano-DESI - HRMS(+) (%) | Peak Abundance nano-DESI-HRMS(-) (%) | Peak Abundance DART-MS(+) (%) | Peak Abundance DART-MS(-) (%) | Tentative Assignment |
|---|---|---|---|---|---|---|---|---|
| Monomers | 121 | $C_7H_7ON$ | $H^+$ | 0.88 | 11 | 4.0 | 20 | 2-formylformanilide |
| | 130 | - | - | - | - | 7.7 | 0.03 | |
| | 131 | $C_8H_5ON$ | $H^+$ | 2.7 | - | 9.0 | 0.10 | 3-oxyindole |
| | 133 | $C_8H_7ON$ | $H^+$ | 0.65 | 0.10 | 4.8 | 0.65 | indoxyl, 3-oxindole |
| | 137 | $C_7H_7O_2N$ | - | - | 8.2 | 4.1 | 9.1 | anthranilic acid |
| | 146 | $C_8H_6ON_2$ | $H^+$ | 0.47 | - | 13 | 1.4 | |
| | 147 | $C_8H_5O_2N$ | $H^+$, $Na^+$ | 0.79, 64 | 11 | 34 | 23 | isatin |
| | 162 | $C_8H_6O_2N_2$ | - | - | 1.8 | 3.2 | 28 | |
| | 163 | $C_8H_5O_3N$ | $Na^+$ | 6.7 | 17 | 16 | 62 | isatoic anhydride |
| | 165 | $C_8H_7O_3N$ | $Na^+$ | 2.3 | 100 | 3.1 | 100 | |
| | 181 | $C_8H_7O_4N$ | - | - | 8.9 | 0.92 | 16 | |
| Dimers | 246 | $C_{16}H_{10}ON_2$ | - | - | 0.37 | 43 | 5.0 | indoxyl red |
| | 248 | $C_{15}H_8O_2N_2$ | $H^+$, $Na^+$ | 3.2, 30 | - | 53 | 1.6 | |
| | 250 | $C_{15}H_{10}O_2N_2$ | $H^+$, $Na^+$ | 36, 21 | 1.0 | 100 | 6.1 | tryptanthrin |
| | 252 | $C_{15}H_{12}O_2N_2$ | $H^+$, $Na^+$ | 75, 0.67 | 0.72 | 19 | 4.6 | |
| | 262 | $C_{16}H_{10}O_2N_2$ | - | - | 0.88 | 58 | 11 | indirubin, indigo dye |
| | 264 | $C_{16}H_{12}O_2N_2$ | $H^+$, $Na^+$ | 7.8, 2.4 | 0.36 | 77 | 12 | dihydro indigo dye |
| | 266 | $C_{15}H_{10}O_3N_2$ | $H^+$, $Na^+$ | 4.9, 21 | 8.9 | 45 | 14 | |
| | 280 | $C_{16}H_{12}O_3N_2$ | $H^+$, $Na^+$ | 17, 44 | 2.8 | 38 | 11 | |
| | 282 | $C_{15}H_{10}O_4N_2$ | $H^+$, $Na^+$ | 0.26, 1.3 | 5.7 | 9.7 | 14 | |
| | 282 | $C_{16}H_{14}O_3N_2$ | $H^+$, $Na^+$ | 100, 0.29 | 0.34 | | | |
| | 294 | $C_{16}H_{10}O_4N_2$ | $H^+$, $Na^+$ | 0.41, 26 | 2.1 | 18 | 12 | |
| | 296 | $C_{16}H_{12}O_4N_2$ | $H^+$, $Na^+$ | 3.7, 48 | 3.8 | 11 | 9.4 | |
| | 310 | $C_{16}H_{10}O_5N_2$ | $Na^+$ | 2.2 | 3.4 | 3.7 | 8.7 | |
| | 312 | $C_{16}H_{12}O_5N_2$ | $H^+$, $Na^+$ | 0.18, 5.6 | 4.3 | 2.0 | 7.4 | |

