# Peer review of "Secondary Organic Aerosol from Atmospheric Photooxidation of Indole"

_Atmospheric Chemistry and Physics, 2017_

## Referee Comment (RC1) · Anonymous Referee #1 · 26 Apr 2017

This manuscript describes a laboratory study on the photo-oxidation of indole under low-NOx conditions. Secondary organic aerosols (SOA) were chemically characterize to determine the importance of indole chemistry in the formation of brown carbon (BrC) constituents. State of art analytical techniques (high-resolution mass spectrometers) used is this study, provide novel and important insights into the understanding of the formation of BrC in the atmosphere. In addition, the authors have evaluated the importance of their findings using a regional model and have highlighted that oxidation of indole could have an important role in SOA formation as well as BrC compounds. Since these results help to provide accurate atmospheric chemistry models for the oxidation of BVOC, they are an important contribution to the literature. While the results are interesting and are appropriate for Atmospheric Chemistry and Physics, few clarifications (c.f. comments below) should be provided and would benefit from clarifying

revisions.

Page 2. Lines 7-15: The authors should consider adding some information on the emissions of indole in the atmosphere and compare them with the emissions of other BVOCs. In addition, what is the contribution of the human activities (e.g. agriculture, pharmaceutical application) in the global emission of indole?

Page 4. Line 5: Could the authors provide more information on the settings of the PTR as well as the time and mass resolutions?

Line 7. The authors mentioned Page 7, line 7 that "The particles had a geometric mean diameter of approximately 0.3 $\mu$m". What was the mean diameter of the seed aerosol?

Line 9. Hallquist et al. is not an appropriate reference. It is a review and they have not determined any aerosol density. Please provide a better reference(s).

The authors decided to use a density of 1.2 g cm-3, could they explain why? Previous studies have reported a density of $\sim$ 1.4 g cm-3 for SOA generated from the oxidation of mono- and polyaromatic compounds, such as naphthalene (Ng et al., 2007; Chan et al., 2009; Chen et al., 2016). It is worth noting that such compounds could form oligomers as proposed in the manuscript (Healy et al., 2012). Could the authors further support their choice and discuss the potential impact of using a density of 1.4 g cm-3 in their model?

The authors haven't discussed the potential wall losses of organic vapors. Have they considered them in the determination of the SOA yields? If not, by looking at the decay of the main products they should be able to provide an estimation.

Lines 14-16: The authors mentioned page 4 lines 2-3: "In some experiments, mixing was not complete by the time the lamps were turned on as evidenced by the measured indole concentrations continuing to increase". How did they determine the SOA yields in such experiments?

[Figure]

Lines 27-28: The time series of the species presented in Figure S2.5 is not clear. The products are continuously produced over the course of the experiments (c.f. formic acid) disregarding the presence of OH radicals (i.e. lights On or Off). The authors should further discuss the time evolution of the identified products.

In addition, the authors proposed that the oxidation of indole form 3-oxyindole. But according to Figure S2.4, this product is an impurity from the indole and it is not formed during the oxidation process (no increase after the lamps were turned on). The decay of 3-oxydinole is much slower than the decay of indole. Therefore, if the 3-oxydinole was really formed from the oxidation of indole it should have shown up, such as 2-formylformanilide. Please clarify.

Lines 19-22: It is not clear if the authors claim that the formation of the oligomers (BrC compounds) occur in the gas phase. The authors used a reference that described similar oxidation process occurring in the liquid phase. If the authors expect that the formation of oligomers happened in the gas phase, they need to provide further evidence and in general better discuss the formation of the identified products. I would suspect this reaction to occur in the particle phase rather than in the gas phase. Have the authors performed any experiments at different RH and/or without seed aerosols? Further experiments are needed if they want to conclude that.

Lines 23-24: With the data presented in Figures S2 it seems difficult to propose that 3-oxyindole is directly formed from the oxidation of indole (c.f. previous comments). Moreover, the authors should consider the mechanism proposed by Healy et al. 2012. In this previous study, they have reported the formation of dimers and oligomers from the photolysis of nitro-naphthalene through gas-phase processes. Could such chemical pathways contribute to the formation of some compounds identified in this manuscript?

---

## Referee Comment (RC2) · Anonymous Referee #2 · 26 Apr 2017

General comments

The paper reports secondary organic aerosol (SOA) yield, mass absorption coefficient (MAC) values, and light-absorbing compounds in SOA produced from photooxidation of indole. High SOA yield was observed and several compounds (i.e. -heterocyclics) were found to absorb visible light. Additionally, an air quality simulation was used to evaluate the contribution of indole-SOA to total atmospheric aerosol in South Coast Air Basin of California (SoCAB) area. The estimated indole-SOA mass loading in SoCAB was low. Nevertheless, due to its high MAC values and SOA yield, indole-SOA has potential to degrade visibility during springtime flowering events in areas with less black carbon influence. The topic of this study fits within the scope of the Atmospheric Chemistry and Physics journal. Also, the SOA formation indole photooxidation as demonstrated by this study can improve SOA model and reduce the gap between model and measurement.

The paper is written well and there are only a few typing errors. I recommend publishing the paper after minor revisions as detailed in the following.

Specific comments

1. Introduction Emission factor or mixing ratio of indole should be discussed to provide a context of the importance of studying SOA formation from indole. Thus we can compare whether indole is as atmospherically relevant as other biogenic VOCs, for example, monoterpenes and isoprene.

2. Methods

a. Pg. 4 Lns. 28-31: is solubility of the light-absorbing compounds similar in both acetonitrile and methanol? For example, isatin has been shown to have different solubility in a set of organic solvents (Liu et al., 2014). Would different solubility affect your results and discussion?

b. Pg. 5 Ln. 7: are the additional samples coming from the same set of experiments or from different experiments with similar conditions? Please clarify.

3. Results and discussion

Fig. S2.3: 2-formylformanilide and isatin rose slightly after injection of indole before oxidation, and the increase were intensified after the chamber lamps were turned on. Is it possible that it was formed by reaction of indole with existing OH in the chamber? How clean was the chamber prior each experiment?

Technical comments

Pg. 8 Ln. 22: Delete "#3725"

Pg. 9 Ln. 19: . . ..MW = 262 Da)

Pg. 10 Lns. 20-21: . . .dihydro. . .

Pg. 11 Lns. 12-14: A diurnal profile of indole would be useful here.

Fig. 8: Caption "...dihydro indigo dye..."

References

Jin-Qiang Liu, Si-Yu Chen, and Baoming Ji, Solubility and Thermodynamic Functions of Isatin in Pure Solvents, Journal of Chemical & Engineering Data, 2014, 59 (11), p.3407-3414, doi: 10.1021/je500396b

---

## Referee Comment (RC3) · Anonymous Referee #3 · 26 Apr 2017

Indole is emitted from the biosphere by plants under stressed conditions. The SOA yield of Indole was measured in a smog chamber. The particles were collected and the mass absorption coefficient determined at various wavelengths. With nanospray desorption electrospray high resolution mass spectrometry and DART-MS the molecular formula of a series of species in the aerosol phase was identified and attributed to possible compounds. Some of these were confirmed by their UV spectra with HPLC-PDA-HRMS. Since many of the products determined absorb in the UV, the authors hypothesize, that indole SOA may considerably contribute to brown carbon. They investigate this with the help of an airshed model, which they updated with some new indole oxidation reactions. They conclude that indole SOA can considerably contribute to decreased visibility and poor air quality in rural and agricultural areas. The paper is well presented. The experimental work is well done and adds new information to a

potentially important, but largely unexplored field of biogenic emissions. However, the interpretation or speculation of chemical mechanisms on product formation and the potential impact on brown carbon are in my view not very solid. The authors propose that the dimer dihydro indigo dye is formed by recombination of two alkyl radicals. At ambient concentrations of indole oxygen would add on much faster than such a recombination of alkyl radicals. The authors may estimate if this mechanism is at all possible at the high concentrations of the experiments. In case of dihydro indoxyl red formation the authors suggest a reaction of the 3-oxindole alkyl radical with indole. As indole is present at really high concentrations this might be an option. However, in both cases such reactions might only be possible in their smog chamber due to the high concentrations. They are most probably not relevant at all for the ambient atmosphere. In Figure 9b, do the authors believe that anthranilic acid and isatin react in the gas phase to tryptanthrin? Such complex non-radical reactions are very slow. Similarly, the oxidation of isatin is formally the addition of two OH radicals or H2O2. Mechanistically, it is quite difficult to imagine this happening in the gas phase. Novotna did the experiments in dichloromethane solution, a fairly different chemical environment. Both of these proposed mechanisms are very speculative and would need further support by literature data or experiments. As already mentioned many products might only have been formed due to the high concentrations used in these experiments. This makes it difficult to extrapolate the results to the real atmosphere. Furthermore, the airshed model includes a reaction of indole to indigo dye, which then partitions into the aerosol. As far as I understand all reacted indole ends up in indigo dye and contributes to SOA. This is a large overestimation. Indigo dye, which is still very reactive, does also not further react. This is unrealistic and all this leads to a large overestimation of the brown carbon effect. It is not very likely that the SOA from indole has finally such a low degree of oxidation at the modelled aerosol concentrations. The reaction time or OH exposure in these experiments was rather low, only 2-3 hours of ambient background OH exposure. Further oxidation reactions would break the chromophore at some point and decrease the brown carbon effect. The paper does not convincingly demonstrate that

the species measured and included in the model are relevant for the ambient. Thus, the paper should include also measurements at lower concentrations and higher OH exposure to demonstrate their relevance.

Minor comments: Page 6, line 17: delete "to the" Page 7, line 32: Figure S2.3 instead of S2.2 Page 8, line 4: suggesting that... (delete "the") Page 8, line 12: the losses should be seen in the PTR-MS Page 8, line 29: spectrometer "of" Page 10, line 31: should be Figure 7 Figure 7: did the authors check the retention time with authentic standards? This would strongly support their assignment Figure 8: replace ingigo by indigo Figure S2.1: Why does indole still decrease after lights off? Figure S2.5: m/z 98 and 99 do already continuously increase before light on? Is there really additional formation when light is turned on?

---

## Referee Comment (RC4) · Anonymous Referee #4 · 26 Apr 2017

Review of Montoya et al. 2017:

Montoya et al. investigate secondary organic aerosol (SOA) formation from OH oxidation of indole. They clearly demonstrate that this typical heterocyclic nitrogen-containing compound is an effective SOA precursor with a yield of ~1. SOA from indole strongly absorb sunlight in the UV region, contributing to brown carbon in the atmosphere. This comprehensive study is important for understanding SOA formation, and the manuscript should be published in Atmospheric Chemistry and Physics (ACP), after considering the individual comments below.

(1) Introduction: Indole can also be emitted from animal husbandry (Feilberg et al., 2010; Yuan et al., 2017). Previous study has showed that animal feeding facilities in Los Angeles areas can be an important emission source for many pollutants (e.g. ammonia) (Nowak et al., 2012). The emission from animal feeding should be included in the discussion and in the chemical transport model as well.

(2) P8 L22: Please correct this citation

(3) P9 L16: Isatoic anhydride should be C8H5O3N

(4) P8-P9: I would suggest moving Figure S2.3 and Figure S3 to the main text. These two graphs are really important to understand the oxidation chemistry of indole.

(5) P11: The reaction of NO3 and indole is not investigated and considered here in this paper. Could you provide some discussion on this. What if you assume NO3 oxidation of indole has a similar SOA yield as photooxidation in the chemical transport model?

(6) Figure 3: Do you consider the wall loss of semi-volatile organic compounds for the chamber experiments.

(7) Figure 5: Could you indicate the locations of the important secondary products in the mass spectra.

(8) Figure 7: Could you add reference lines for the labelled compound names.

(9) Figure 8: Could you provide a colored version of this graph.

(10) Figure 9: Could you combine (a) and (b) to provide a more combined mechanism for the reactions. Based on Atkinson et al. 1995, 2-formyl-formanilide is the large oxidation product of OH+indole. This information needs to reflect in Figure 9.

**References:**
Feilberg, A., Liu, D., Adamsen, A. P. S., Hansen, M. J., and Jonassen, K. E. N.: Odorant Emissions from Intensive Pig Production Measured by Online Proton-Transfer-Reaction Mass Spectrometry, Environmental Science & Technology, 44, 5894-5900, 10.1021/es100483s, 2010.
Nowak, J. B., Neuman, J. A., Bahreini, R., Middlebrook, A. M., Holloway, J. S., McKeen, S. A., Parrish, D. D., Ryerson, T. B., and Trainer, M.: Ammonia sources in the California South Coast Air Basin and their impact on ammonium nitrate formation, Geophysical Research Letters, 39, L07804, 10.1029/2012GL051197, 2012.
Yuan, B., Coggon, M. M., Koss, A. R., Warneke, C., Eilerman, S., Peischl, J., Aikiin, K. C., Ryerson, T. B., and de Gouw, J. A.: Emissions of volatile organic compounds (VOCs) from concentrated animal feeding operations (CAFOs): chemical compositions and separation of sources, Atmos. Chem. Phys., 17, 4945-4956, 10.5194/acp-17-4945-2017, 2017.

---

## Author Comment (AC1) · 16 Jul 2017

See attached.
* * *
[Figure]

**Response to Anonymous Referee #1**

This manuscript describes a laboratory study on the photo-oxidation of indole under low-NOx conditions. Secondary organic aerosols (SOA) were chemically characterized to determine the importance of indole chemistry in the formation of brown carbon (BrC) constituents. State of art analytical techniques (high-resolution mass spectrometers) used is this study, provide novel and important insights into the understanding of the formation of BrC in the atmosphere. In addition, the authors have evaluated the importance of their findings using a regional model and have highlighted that oxidation of indole could have an important role in SOA formation as well as BrC compounds. Since these results help to provide accurate atmospheric chemistry models for the oxidation of BVOC, they are an important contribution to the literature. While the results are interesting and are appropriate for Atmospheric Chemistry and Physics, few clarifications (c.f. comments below) should be provided and would benefit from clarifying revisions.

**1.1** Page 2. Lines 7-15: The authors should consider adding some information on the emissions of indole in the atmosphere and compare them with the emissions of other BVOCs. In addition, what is the contribution of the human activities (e.g. agriculture, pharmaceutical application) in the global emission of indole?

The literature review on the emission sources of indole that was included in the introduction section manuscript was fairly comprehensive. However, we have added additional references dealing with emissions of indole from animal husbandry (see response 4.1 to reviewer #4).

**1.2** Page 4. Line 5: Could the authors provide more information on the settings of the PTR as well as the time and mass resolutions?

We added information on the PTR-ToF-MS settings to the first paragraph in section 2.

**Secondary Organic Aerosol from Atmospheric Photooxidation of Indole**

Julia Montoya-Aguilera,[1] Jeremy R. Horne,[2] Mallory L. Hinks,[1] Lauren T. Fleming,[1] Véronique Perraud,[1] Peng Lin,[3] Alexander Laskin,[3] Julia Laskin,[34] Donald Dabdub,[2] and Sergey A. Nizkorodov[1]

[1]Department of Chemistry, University of California, Irvine, CA 92697, USA
[2]Department of Mechanical and Aerospace Engineering, University of California, Irvine, CA 92697, USA
[3]Environmental Molecular Sciences Laboratory, Pacific Northwest National Laboratory, Richland, WA 99354, USA
[4]Physical Sciences Division Pacific Northwest National Laboratory, Richland, WA 99354, USA
[3]Department of Chemistry, Purdue University, West Lafayette, IN 47907, USA

*Correspondence to*: Sergey A. Nizkorodov (nizkorod@uci.edu)

**Abstract.** Indole is a heterocyclic compound emitted by various plant species under stressed conditions or during flowering events. The formation, optical properties, and chemical composition of secondary organic aerosol (SOA) formed by low-$NO_x$ photooxidation of indole were investigated. The SOA yield (1.1 3 ± 0.3) was estimated from measuring the particle mass concentration with a scanning mobility particle sizer (SMPS) and correcting it for the wall loss effects. The high value of the SOA mass yield suggests that the majority of most oxidized indole products eventually end up in the particle phase. The SOA particles were collected on filters and analysed offline with UV-Vis spectrophotometry to measure the mass absorption coefficient (MAC) of the bulk sample. The samples were visibly brown and had MAC values of ~2 $m^2$/g at $\lambda =$ 300 nm and ~0.5 $m^2$/g at $\lambda = 400$ nm, comparable to strongly absorbing brown carbon emitted from biomass burning. The

Comment [SN1]: The last name of the first author has changed between the ACPD and A submissions. Congratulations Julia!

---

## Author Comment (AC2) · 16 Jul 2017

**Response to Anonymous Referee #2**

General comments

The paper reports secondary organic aerosol (SOA) yield, mass absorption coefficient (MAC) values, and light-absorbing compounds in SOA produced from photooxidation of indole. High SOA yield was observed and several compounds (i.e. -heterocyclics) were found to absorb visible light. Additionally, an air quality simulation was used to evaluate the contribution of indole-SOA to total atmospheric aerosol in South Coast Air Basin of California (SoCAB) area. The estimated indole-SOA mass loading in SoCAB was low. Nevertheless, due to its high MAC values and SOA yield, indole-SOA has potential to degrade visibility during springtime flowering events in areas with less black carbon influence. The topic of this study fits within the scope of the Atmospheric Chemistry and Physics journal. Also, the SOA formation indole photooxidation as demonstrated by this study can improve SOA model and reduce the gap between model and measurement.

The paper is written well and there are only a few typing errors. I recommend publishing the paper after minor revisions as detailed in the following. Specific comments:

**2.1** Emission factor or mixing ratio of indole should be discussed to provide a context of the importance of studying SOA formation from indole. Thus we can compare whether indole is as atmospherically relevant as other biogenic VOCs, for example, monoterpenes and isoprene.

In addition to the literature review on the emission sources of indole that was already included in the introduction section, we have added additional references dealing with emissions of indole from animal husbandry (see response 4.1 to reviewer #4).

**2.2** Pg. 4 Lns. 28-31: is solubility of the light-absorbing compounds similar in both acetonitrile and methanol? For example, isatin has been shown to have different solubility in a set of organic solvents (Liu et al., 2014). Would different solubility affect your results and discussion?

Based on visual inspection, the samples appeared to dissolve fully in both acetonitrile and methanol. If we use isatin as an example, then our solutions are at least 80 times more dilute than the molar solubilities reported for acetonitrile (Liu et al., 2014) and methanol (Baluja et al., 2013). More clarification was added in section 2, paragraphs 3-4.

**2.3** Pg. 5 Ln. 7: are the additional samples coming from the same set of experiments or from different experiments with similar conditions? Please clarify.

More detail was added in the experimental methods section to clarify that additional samples were obtained from separate experiments run under the same conditions (section 2, paragraph 1).

**2.4** Fig. S2.3: 2-formylformanilide and isatin rose slightly after injection of indole before oxidation, and the increase were intensified after the chamber lamps were turned on. Is it possible that it was formed by reaction of indole with existing OH in the chamber? How clean was the chamber prior each experiment?

It not possible that 2-formylformanilide and isatin formed by an OH reaction before the lamps are turned on, because there is no OH in the chamber without lights. For example, monoterpenes do not get oxidized in our chamber unless lights are on. It is conceivable that indole is more sensitive to oxidation by oxygen or hydrogen peroxide present in the chamber. The SDS (Sigma Aldrich, 2014) for indole

indicates that indole is sensitive to light and air, so it may oxidize to some extent even in the absence of UV lights.

Technical comments

Pg. 8 Ln. 22: Delete "#3725"

Pg. 9 Ln. 19: . . ..MW = 262 Da)

Pg. 10 Lns. 20-21: . . .dihydro. . .

Fig. 8: Caption ". . .dihydro indigo dye. . ."

All fixed, thank you for pointing out these typos.

Pg. 11 Lns. 12-14: A diurnal profile of indole would be useful here.

Diurnal profiles (Figure S5) have been added to depict the domain wide average concentrations of indole and the domain maximum concentrations of indole.

References

Jin-Qiang Liu, Si-Yu Chen, and Baoming Ji, Solubility and Thermodynamic Functions of Isatin in Pure Solvents, Journal of Chemical & Engineering Data, 2014, 59 (11), p.3407-3414, doi: 10.1021/je500396b

This reference has been added.

---

## Author Comment (AC3) · 16 Jul 2017

**Response to Anonymous Referee #1**

This manuscript describes a laboratory study on the photo-oxidation of indole under low-NOx conditions. Secondary organic aerosols (SOA) were chemically characterized to determine the importance of indole chemistry in the formation of brown carbon (BrC) constituents. State of art analytical techniques (high-resolution mass spectrometers) used is this study, provide novel and important insights into the understanding of the formation of BrC in the atmosphere. In addition, the authors have evaluated the importance of their findings using a regional model and have highlighted that oxidation of indole could have an important role in SOA formation as well as BrC compounds. Since these results help to provide accurate atmospheric chemistry models for the oxidation of BVOC, they are an important contribution to the literature. While the results are interesting and are appropriate for Atmospheric Chemistry and Physics, few clarifications (c.f. comments below) should be provided and would benefit from clarifying revisions.

**1.1** Page 2. Lines 7-15: The authors should consider adding some information on the emissions of indole in the atmosphere and compare them with the emissions of other BVOCs. In addition, what is the contribution of the human activities (e.g. agriculture, pharmaceutical application) in the global emission of indole?

The literature review on the emission sources of indole that was included in the introduction section manuscript was fairly comprehensive. However, we have added additional references dealing with emissions of indole from animal husbandry (see response 4.1 to reviewer #4).

**1.2** Page 4. Line 5: Could the authors provide more information on the settings of the PTR as well as the time and mass resolutions?

We added information on the PTR-ToF-MS settings to the first paragraph in section 2.

**1.3** Line 7. The authors mentioned Page 7, line 7 that "The particles had a geometric mean diameter of approximately 0.3 $\mu$m". What was the mean diameter of the seed aerosol?

No seeds were used in this experiment because the seed material would interfere with HRMS analysis. A clarification has been added to the beginning of section 2.

**1.4** Line 9. Hallquist et al. is not an appropriate reference. It is a review and they have not determined any aerosol density. Please provide a better reference(s). The authors decided to use a density of 1.2 g cm-3, could they explain why? Previous studies have reported a density of ~1.4 g cm-3 for SOA generated from the oxidation of mono- and polyaromatic compounds, such as naphthalene (Ng et al., 2007; Chan et al., 2009; Chen et al., 2016). It is worth noting that such compounds could form oligomers as proposed in the manuscript (Healy et al., 2012). Could the authors further support their choice and discuss the potential impact of using a density of 1.4 g cm-3 in their model?

This is an excellent point. We wish we had tools for measuring aerosol particle density at our disposal. The assumed SOA density was changed from 1.2 to 1.4 g cm$^{-3}$ based on the reported values for naphthalene SOA in Chan et al., 2009 and Chen et al., 2016, and based on densities of known indole oxidation products. This increased the reported aerosol yield by ~15% and decreased the reported MAC by ~15%.

**1.5** The authors haven't discussed the potential wall losses of organic vapors. Have they considered them in the determination of the SOA yields? If not, by looking at the decay of the main products they should be

able to provide an estimation.

We suspect that the wall loss effects are minimal because of the fast aerosol formation and high apparent yield. We added the following statement regarding the possible effect of the wall loss of oxidation products on the yield, "Indole oxidation products could be lost to the walls reducing the apparent yield and contributing to its scatter. However, this effect is probably minor given that the apparent yield is quite high." in section 4.1, paragraph 2.

**1.6** Lines 14-16: The authors mentioned page 4 lines 2-3: "In some experiments, mixing was not complete by the time the lamps were turned on as evidenced by the measured indole concentrations continuing to increase". How did they determine the SOA yields in such experiments?

In all yield calculations, we relied on the actual amount of injected indole so we do not view incomplete mixing at the state of the reaction as a huge problem. As long as mixing timescale is shorter than the oxidation timescale (which is the case) the yield calculation should work reasonably well. We added the following statement, "Although mixing was not fast, it was faster than the time scale of the reaction, so it should not have affected the SOA mass yield calculations." to the first paragraph in section 2.

**1.7** Page 7. Lines 27-28: The time series of the species presented in Figure S2.5 is not clear. The products are continuously produced over the course of the experiments (c.f. formic acid) disregarding the presence of OH radicals (i.e. lights On or Off). The authors should further discuss the time evolution of the identified products. In addition, the authors proposed that the oxidation of indole form 3-oxyindole. But according to Figure S2.4, this product is an impurity from the indole and it is not formed during the oxidation process (no increase after the lamps were turned on). The decay of 3-oxydinole is much slower than the decay of indole. Therefore, if the 3-oxydinole was really formed from the oxidation of indole it should have shown up, such as 2- formylformanilide. Please clarify.

We found the time dependence of some of the observed ion abundances puzzling. After some more experiments with PTR-ToF-MS we suspect that some of the behavior results from slow-adsorption-desorption kinetics on the sampling lines. The PTR-ToF-MS flexible heated inlet is not long enough to reach into the chamber, and we have to sample through segments of unheated Teflon tubing. Stickier compounds take longer to pass through the tubing to the PTR-ToF-MS instrument, and also compete with other compounds for surface sites. This complicates the interpretation. In response to this comment, as well as related comments from other reviewers we did the following changes:

- better explained the limitations of PTR-ToF-MS measurements in the supporting information section
- removed a statement that the time dependence of isatin implied a complex mechanism of its production
- removed Figure S2.5

**1.8** Page 10. Lines 19-22: It is not clear if the authors claim that the formation of the oligomers (BrC compounds) occur in the gas phase. The authors used a reference that described similar oxidation process occurring in the liquid phase. If the authors expect that the formation of oligomers happened in the gas phase, they need suspect this reaction to occur in the particle phase rather than in the gas phase. Have the authors performed any experiments at different RH and/or without seed aerosols? Further experiments are needed if they want to conclude that.

We have added a new paragraph at the end of section 4.1 to address this point. The added discussion emphasizes the tentative nature of the proposed mechanism.

We agree that multiple further experiments are needed to clarify the mechanism and establish where and how different processes take place. Our focus was not a complete analysis of the mechanism – this would take a prohibitively long time, well beyond a lifetime of a typical graduate student. We hope this publication inspires additional work on this interesting SOA system.

**1.9** Lines 23-24: With the data presented in Figures S2 it seems difficult to propose that 3-oxyindole is directly formed from the oxidation of indole (c.f. previous comments).

We agree that the fact that 3-oxyindole is present as an impurity complicates the interpretation. However, it appears to be also a product of indole photooxidation. To make it easier to see we replotted the figure on a linear scale in this response. The "bulge" observed at ~130 minutes would not be there if 3-oxyindole was only being removed.

[Figure]

*In the graph above, the indole peak abundance was scaled to 1/50 of its actual peak abundance.

**1.10** Moreover, the authors should consider the mechanism proposed by Healy et al. 2012. In this previous study, they have reported the formation of dimers and oligomers from the photolysis of nitro-naphthalene through gas-phase processes. Could such chemical pathways contribute to the formation of some compounds identified in this manuscript?

Thank you for this suggestion. We have added the following to the discussion of the mechanism in section 4.1, paragraph 11: "For example, Healy et al. (2012) observed efficient dimerization of naphthoxy radicals in the gas phase leading to rapid formation of SOA following photolysis of 1-nitronaphthalene. The dimerization of oxindole to dihydro indigo dye, as well as other oligomerization processes in indole SOA, could follow a mechanism similar to the one described by Healy et al. (2012)."

---

## Author Comment (AC4) · 16 Jul 2017

**Response to Anonymous Referee #3**

Indole is emitted from the biosphere by plants under stressed conditions. The SOA yield of Indole was measured in a smog chamber. The particles were collected and the mass absorption coefficient determined at various wavelengths. With nanospray desorption electrospray high resolution mass spectrometry and DART-MS the molecular formula of a series of species in the aerosol phase was identified and attributed to possible compounds. Some of these were confirmed by their UV spectra with HPLC-PDA-HRMS. Since many of the products determined absorb in the UV, the authors hypothesize, that indole SOA may considerably contribute to brown carbon. They investigate this with the help of an airshed model, which they updated with some new indole oxidation reactions. They conclude that indole SOA can considerably contribute to decreased visibility and poor air quality in rural and agricultural areas. The paper is well presented. The experimental work is well done and adds new information to a potentially important, but largely unexplored field of biogenic emissions. However, the interpretation or speculation of chemical mechanisms on product formation and the potential impact on brown carbon are in my view not very solid.

**3.1** The authors propose that the dimer dihydro indigo dye is formed by recombination of two alkyl radicals. At ambient concentrations of indole oxygen would add on much faster than such a recombination of alkyl radicals. The authors may estimate if this mechanism is at all possible at the high concentrations of the experiments. In case of dihydro indoxyl red formation the authors suggest a reaction of the 3-oxindole alkyl radical with indole. As indole is present at really high concentrations this might be an option. However, in both cases such reactions might only be possible in their smog chamber due to the high concentrations. They are most probably not relevant at all for the ambient atmosphere.

This is an excellent point. We have revised Figure 9 to eliminate reactions that would require recombination of two carbon-centered radicals. We have also added a reference to Healy et al. (2012) that discussed a related mechanism of oligomerization in naphthalene oxidation, which was interpreted by efficient reaction of two resonance-stabilized free radicals. Finally, we added a paragraph to the end of section 4.1 intended to emphasize the tentative nature of the outlined mechanism.

**3.2** In Figure 9b, do the authors believe that anthranilic acid and isatin react in the gas phase to tryptanthrin? Such complex non-radical reactions are very slow.

Indeed, the reaction would be very slow in the gas phase. We therefore suspect it should occur in the particle-phase. We have added a statement in the paper suggesting that some of the products may result from particle-phase chemistry (even though we cannot directly prove it with our data). The new text appears in the last paragraph of section 4.1.

**3.3** Similarly, the oxidation of isatin is formally the addition of two OH radicals or H2O2. Mechanistically, it is quite difficult to imagine this happening in the gas phase. Novotna did the experiments in dichloromethane solution, a fairly different chemical environment. Both of these proposed mechanisms are very speculative and would need further support by literature data or experiments. As already mentioned many products might only have been formed due to the high concentrations used in these experiments. This makes it difficult to extrapolate the results to the real atmosphere.

Indeed, the processes in the Novotna et al. experiments in dichloromethane took longer time compared to the timescale of chamber reaction in aerosol, and it is not obvious whether they would occur any faster under chamber conditions. We therefore view the proposed explanation for the formation of this particular chromophore as tentative, and acknowledge this in the revised version.

**3.4** Furthermore, the airshed model includes a reaction of indole to indigo dye, which then partitions into the aerosol. As far as I understand all reacted indole ends up in indigo dye and contributes to SOA. This is a large overestimation. Indigo dye, which is still very reactive, does also not further react. This is unrealistic and all this leads to a large overestimation of the brown carbon effect. It is not very likely that the SOA from indole has finally such a low degree of oxidation at the modelled aerosol concentrations. The reaction time or OH exposure in these experiments was rather low, only 2-3 hours of ambient background OH exposure. Further oxidation reactions would break the chromophore at some point and decrease the brown carbon effect. The paper does not convincingly demonstrate that the species measured and included in the model are relevant for the ambient. Thus, the paper should include also measurements at lower concentrations and higher OH exposure to demonstrate their relevance.

We have added a statement in the paper that we hope better explains that the choice of the surrogate oxidation product is not too important for this system because most of the products have fairly low volatility. Any of the choices of the surrogate product would suffer from the same issue the reviewer is referring to – many of them can be expected to react further via various aging mechanisms. The indole SOA will indeed continue to age, likely in a very interesting way. We have not explored OH-driven aging or photolysis-driven aging of indole SOA in this study but we fully intend to do so in a follow up work. At present time it is hard for us to estimate the effect of the increased OH exposure but we are getting an oxidation flow reactor soon that will make such experiments possible.

We also agree that experiments at lower concentrations would be more environmentally relevant. The choice of concentration for this initial study was high because we were concerned about getting sufficient signal-to-noise ratio in mass spectrometry and UV/Vis measurements. Now that we know what to expect we can move to better designed experiments at lower concentrations.

Minor comments:

Page 6, line 17: delete "to the"

Page 7, line 32: Figure S2.3 instead of S2.2

Page 8, line 4: suggesting that*. . .* (delete "the")

Page 8, line 29: spectrometer "of"

Page 10, line 31: should be Figure 7

All of these corrections have been made.

Page 8, line 12: the losses should be seen in the PTR-MS

We have reworded the SOA yield discussion to avoid ambiguities

Figure 7: did the authors check the retention time with authentic standards? This would strongly support their assignment

We were not able to get authentic standards in time for this experimental campaign. Fortunately, the optical absorption spectra are sufficiently complex and coincidence in peaks in the absorption spectra with the corresponding peaks of the reported standards is in our opinion quite convincing.

Figure 8: replace ingigo by indigo

Corrected

Figure S2.1: Why does indole still decrease after lights off?

When we collect the sample we allow make up air to enter the chamber and dilute its content. The indole

would not decrease if we were not collecting. We added a note to the caption of Figure S2.1.

Figure S2.5: m/z 98 and 99 do already continuously increase before light on? Is there really additional formation when light is turned on?
We have elected to remove Figure S2.5 from the SI section and focus on larger PTR-ToF-MS signals, which have less ambiguous interpretation.

---

## Author Comment (AC5) · 16 Jul 2017

**Response to Anonymous Referee #4**

Montoya et al. investigate secondary organic aerosol (SOA) formation from OH oxidation of indole. They clearly demonstrate that this typical heterocyclic nitrogen-containing compound is an effective SOA precursor with a yield of ~1. SOA from indole strongly absorb sunlight in the UV region, contributing to brown carbon in the atmosphere. This comprehensive study is important for understanding SOA formation, and the manuscript should be published in Atmospheric Chemistry and Physics (ACP), after considering the individual comments below.

**4.1** Introduction: Indole can also be emitted from animal husbandry (Feilberg et al., 2010; Yuan et al., 2017). Previous study has showed that animal feeding facilities in Los Angeles areas can be an important emission source for many pollutants (e.g. ammonia) (Nowak et al., 2012). The emission from animal feeding should be included in the discussion and in the chemical transport model as well. This is a good point. We added the following paragraph in the introduction: "An additional potential source of indole is animal husbandry, however, the emission rate of indole from this source remains uncertain. In concentrated animal feeding operations (CAFOs), indole is primarily emitted from animal waste (Yuan et al., 2017), and can contribute significantly to the malodors in cattle feedyards and swine facilities (Feilberg et al., 2010; Wright et al., 2005). While Yuan et al. (2017) indicated that indole is emitted from dairy operations, beef feedyards, sheep feedyards, and chicken feedyards, the emission rate for indole from these sources was not quantified. Other studies have quantified the emission rate for indole, but only for pig facilities (Feilberg et al., 2010; Hobbs et al., 2004). The United States Department of Agriculture (USDA) 2012 census agriculture atlas maps show no hogs or pigs in the model domain used in this study. Furthermore, Hobbs et al. (2004) showed only trace emissions of indole from cattle slurry, and did not detect indole from laying hen manure. Thus, emissions of indole from animal husbandry are not included in this study, but should be considered when modelling areas with active pig facilities."

**4.2** P8 L22: Please correct this citation
Citation was corrected.

**4.3** P9 L16: Isatoic anhydride should be C8H5O3N
Formula was corrected.

**4.4** P8-P9: I would suggest moving Figure S2.3 and Figure S3 to the main text. These two graphs are really important to understand the oxidation chemistry of indole.
We elected to keep these figures in the SI for two reasons. Firstly, we have too many figures already. Secondly, the interpretation of these figures is complicated somewhat by the impurities present in the indole samples we used, as well as adsorption of sticky indole oxidation products to the PTR-ToF-MS sampling lines. While these technical complications do not change the main conclusion of the paper in a critical way, they would be too distracting to the readers in the main text. We think the SI section is an appropriate place for these figures under the circumstances.

**4.5** P11: The reaction of NO3 and indole is not investigated and considered here in this paper. Could you

provide some discussion on this. What if you assume NO3 oxidation of indole has a similar SOA yield as photooxidation in the chemical transport model?

Inclusion of the reaction of $NO_3$ and indole in the simulations is a good idea. It is possible that it makes a non-negligible amount of SOA given that the rate constant for the $NO_3$+indole reaction is the same order of magnitude as the OH+indole reaction. However, we elected not to include the $NO_3$+indole reaction in this study because we have no experimental information on the yield or chemical composition of SOA produced via the $NO_3$+indole reaction. We are interested in exploring this reaction in a follow up study with corresponding laboratory experiments - thank you for the suggestion.

**4.6** Figure 3: Do you consider the wall loss of semi-volatile organic compounds for the chamber experiments.

Based on the high value of the SOA mass yield, and the fast rate of production of particles in the chamber, the wall effects must be minimal. We added the following statement in the paper to address this: "Indole oxidation products could be lost to the walls reducing the apparent yield and contributing to its scatter. However, this effect is probably minor given that the apparent yield is quite high." in section 4.1, paragraph 2.

**4.7** Figure 5: Could you indicate the locations of the important secondary products in the mass spectra.

The five most abundant peaks in each mass spectrum are now denoted in the updated figure.

**4.8** Figure 7: Could you add reference lines for the labelled compound names.

Reference lines were added for the bold-face assignments.

**4.9** Figure 8: Could you provide a colored version of this graph.

The reference spectra have been changed from black to blue.

**4.10** Figure 9: Could you combine (a) and (b) to provide a more combined mechanism for the reactions. Based on Atkinson et al. 1995, 2-formyl-formanilide is the large oxidation product of OH+indole. This information needs to reflect in Figure 9.

We are not trying to provide a comprehensive mechanism in Figure 9; its main purpose is to propose possible formation mechanisms of the light-absorbing compounds. We did attempt to merge parts a and b, but it made the figure too cluttered. However, adding 2-formyl-formanilide is a good suggestion. The figure has been modified to include it.

**References:**
Feilberg, A., Liu, D., Adamsen, A. P. S., Hansen, M. J., and Jonassen, K. E. N.: Odorant Emissions from Intensive Pig Production Measured by Online Proton-Transfer-Reaction Mass Spectrometry, Environmental Science & Technology, 44, 5894-5900, 10.1021/es100483s, 2010.
Nowak, J. B., Neuman, J. A., Bahreini, R., Middlebrook, A. M., Holloway, J. S., McKeen, S. A., Parrish, D. D., Ryerson, T. B., and Trainer, M.: Ammonia sources in the California South Coast Air Basin and their impact on ammonium nitrate formation, Geophysical Research Letters, 39, L07804, 10.1029/2012GL051197, 2012.

Yuan, B., Coggon, M. M., Koss, A. R., Warneke, C., Eilerman, S., Peischl, J., Aikiin, K. C., Ryerson, T. B., and de Gouw, J. A.: Emissions of volatile organic compounds (VOCs) from concentrated animal feeding operations (CAFOs): chemical compositions and separation of sources, Atmos. Chem. Phys., 17, 4945- 4956, 10.5194/acp-17-4945-2017, 2017.

Relevant references from this list have been added, thank you for the suggestion.

---

## Author Comment (AC6) · 16 Jul 2017

[revised manuscript text omitted]

Comment [SN2]: Note to editor: this figure w redrawn during the revisions. Only the revised version is displayed to reduce clutter.

[Figure]

**Figure 4:** Wavelength-dependent mass absorption coefficient (MAC) of indole SOA. The inset shows the log-log version of the same data used to determine the absorption Angstrom exponent (fitted from 300 to 600 nm) as well as photographs of the indole SOA collected on a filter and extracted in methanol.

**Comment [SN3]:** Note to editor: this figure w redrawn during the revisions. Only the revised version is displayed to reduce clutter.

[Figure]

*Note to editors: please use the full page width for figure 5*

Figure 5: nano-DESI and DART mass spectra of indole SOA plotted as a function of the molecular weights of the neutral compounds. The nano-DESI mass spectra contained only peaks assignable to specific formulas, while DART mass spectra peaks contain all observed peaks. The five most abundant peaks in each mass spectrum are indicated with letters: (a) 248 Da, $C_{15}H_8O_2N_2$; (b) 250 Da, $C_{15}H_{10}O_2N_2$, tryptanthrin; (c) 262 Da, $C_{16}H_{10}O_2N_2$, indirubin and/or indigo dye; (d) 264 Da, $C_{16}H_{12}O_2N_2$, dihydro indigo dye; (e) 266 Da, $C_{15}H_{10}O_3N_2$; (f) 147 Da, $C_8H_5O_2N$, isatin; (g) 252 Da, $C_{15}H_{12}O_2N_2$; (h) 280 Da, $C_{16}H_{12}O_3N_2$; (i) 282 Da, $C_{16}H_{14}O_3N_2$; (j) 121 Da, $C_7H_7ON$, 2-formylformanilide; (k) 162 Da, $C_8H_6O_2N_2$; (l) 163 Da, $C_8H_5O_3N_2$, isatoic anhydride (m) 165 Da, $C_8H_7O_3N$.

[Figure]

**Figure 6:** Distribution of the number of C atoms in the indole SOA compounds detected in both positive and negative ion model nano-DESI.

[Figure]

**Figure 7**: HPLC-PDA chromatogram of indole SOA. The absorbance is plotted as a function of both retention time and wavelength. Peaks are labelled by their PDA retention time followed by their proposed assignment. Bold-faced assignments are specific isomers that are discussed further in the text. Note the reference line for dihydro indigo points to a small peak between two larger peaks that obscure it in this projection.

Comment [SN5]: Note to editor: this figure w redrawn during the revisions. Only the revised version is displayed to reduce clutter.

[Figure]

**Figure 8:** Comparison between measured PDA absorption spectra at selected retention time (RTs) and reference spectra of proposed chromophores in the literature (reference spectrum is not available for dihydro  indigo dye, panel e).

**Comment [SN6]:** Note to editor: this figure w redrawn during the revisions. Only the revised version is displayed to reduce clutter.

**(a)**

2-formylformanilide

Indole

3-oxindole

Dihydro indigo dye

+ 3-oxindole, -H

+ indole, -H

Indigo dye

Isatin

Dihydro indoxyl red

Indoxyl red

**(b)**

Indigo dye

Isatin

Isatoic anhydride

Isatin

Anthranilic acid

$CO_2$

Anthranilic acid

Isatin

Tryptanthrin

-2 $H_2O$

**Figure 9:** Tentative mechanism for the formation of observed chromophores in the photooxidation of indole. (a) Processes leading to indigo dye and indoxyl red based on Iddon et al. (1971). (b) Processes leading to tryptanthrin based on Novotna et al. (2003). "Ox" denotes an oxidation step.

**Comment [SN7]:** Note to editor: this figure w redrawn during the revisions. Only the revised version is displayed to reduce clutter.

[Figure]

**Figure 10:** 24-hour average concentrations (µg/m$^3$) of (a) total SOA in the base case and, (b) additional SOA resulting from indole photooxidation in (b)  the low emissions scenario, (c)  the medium emissions scenario, and (d)  the high emission scenario.

[Figure]

**Figure 11:** Domain wide average SOA concentrations in the base case (black line, left axis) and the relative increase in domain wide average SOA concentrations (right axis) due to indole SOA in the (a) low emissions scenario, (b) medium emissions scenario, and (c) high emissions scenario.

**Table 1:** Monomer and dimer peaks with the largest peak abundance observed in DART-MS and nano-DESI-HRMS spectra. Selected peaks corresponding to the compounds shown in Figure 1 are also included. Proposed assignments are based on the formulas from nano-DESI-HRMS. Peak abundances are normalized with respect to the most abundant peak in each spectrum.

| | Nominal Mass | Formula | Ionization by $H^+$ or $Na^+$ in nano-DESI (+) | Peak Abundance nano-DESI - HRMS(+) (%) | Peak Abundance nano-DESI-HRMS(-) (%) | Peak Abundance DART-MS(+) (%) | Peak Abundance DART-MS(-) (%) | Tentative Assignment |
|---|---|---|---|---|---|---|---|---|
| Monomers | 121 | $C_7H_7ON$ | $H^+$ | 0.88 | 11 | 4.0 | 20 | 2-formylformanilide |
| | 130 | - | - | - | - | 7.7 | 0.03 | |
| | 131 | $C_8H_5ON$ | $H^+$ | 2.7 | - | 9.0 | 0.10 | 3-oxyindole |
| | 133 | $C_8H_7ON$ | $H^+$ | 0.65 | 0.10 | 4.8 | 0.65 | indoxyl, 3-oxindole |
| | 137 | $C_7H_7O_2N$ | - | - | 8.2 | 4.1 | 9.1 | anthranilic acid |
| | 146 | $C_8H_6ON_2$ | $H^+$ | 0.47 | - | 13 | 1.4 | |
| | 147 | $C_8H_5O_2N$ | $H^+$, $Na^+$ | 0.79, 64 | 11 | 34 | 23 | isatin |
| | 162 | $C_8H_6O_2N_2$ | - | - | 1.8 | 3.2 | 28 | |
| | 163 | $C_8H_5O_3N$ | $Na^+$ | 6.7 | 17 | 16 | 62 | isatoic anhydride |
| | 165 | $C_8H_7O_3N$ | $Na^+$ | 2.3 | 100 | 3.1 | 100 | |
| | 181 | $C_8H_7O_4N$ | - | - | 8.9 | 0.92 | 16 | |
| Dimers | 246 | $C_{16}H_{10}ON_2$ | - | - | 0.37 | 43 | 5.0 | indoxyl red |
| | 248 | $C_{15}H_8O_2N_2$ | $H^+$, $Na^+$ | 3.2, 30 | - | 53 | 1.6 | |
| | 250 | $C_{15}H_{10}O_2N_2$ | $H^+$, $Na^+$ | 36, 21 | 1.0 | 100 | 6.1 | tryptanthrin |
| | 252 | $C_{15}H_{12}O_2N_2$ | $H^+$, $Na^+$ | 75, 0.67 | 0.72 | 19 | 4.6 | |
| | 262 | $C_{16}H_{10}O_2N_2$ | - | - | 0.88 | 58 | 11 | indirubin, indigo dye |
| | 264 | $C_{16}H_{12}O_2N_2$ | $H^+$, $Na^+$ | 7.8, 2.4 | 0.36 | 77 | 12 | dihydro indigo dye |
| | 266 | $C_{15}H_{10}O_3N_2$ | $H^+$, $Na^+$ | 4.9, 21 | 8.9 | 45 | 14 | |
| | 280 | $C_{16}H_{12}O_3N_2$ | $H^+$, $Na^+$ | 17, 44 | 2.8 | 38 | 11 | |
| | 282 | $C_{15}H_{10}O_4N_2$ | $H^+$, $Na^+$ | 0.26, 1.3 | 5.7 | 9.7 | 14 | |
| | 282 | $C_{16}H_{14}O_3N_2$ | $H^+$, $Na^+$ | 100, 0.29 | 0.34 | | | |
| | 294 | $C_{16}H_{10}O_4N_2$ | $H^+$, $Na^+$ | 0.41, 26 | 2.1 | 18 | 12 | |
| | 296 | $C_{16}H_{12}O_4N_2$ | $H^+$, $Na^+$ | 3.7, 48 | 3.8 | 11 | 9.4 | |
| | 310 | $C_{16}H_{10}O_5N_2$ | $Na^+$ | 2.2 | 3.4 | 3.7 | 8.7 | |
| | 312 | $C_{16}H_{12}O_5N_2$ | $H^+$, $Na^+$ | 0.18, 5.6 | 4.3 | 2.0 | 7.4 | |

---

## Author Response (AR2)

**Response to Anonymous Referee #4**

Most of my concerns from the first-round review have been dealt by the authors. I still think some discussions on indole+NO3 reaction should be included in the manuscript. If this reaction is included, the diurnal profiles of indole concentrations and indole SOA would be totally different (Fig 11 and Fig. S5 in the SI). At this point, the Fig. 11 and Fig. S5 are just misleading.

To explore the potential impact of including the indole+$NO_3$ reaction on modeled gas-phase indole and indole SOA concentrations, an additional scenario was explored using the UCI-CIT model. This scenario corresponds to the high emissions scenario described in the manuscript, with one new gas-phase reaction added to the model's chemical mechanism. For this reaction, it is assumed that gas-phase indole reacts with $NO_3$ to produce indigo dye, the same representative oxidation product as the reaction of gas-phase indole with hydroxyl radical. Therefore, the new gas-phase reaction added into the model forms gas-phase indigo dye via oxidation of gas-phase indole by $NO_3$, using the rate constant of $1.3x10^{-10}$ $cm^3molec^{-1}s^{-1}$ proposed by Atkinson et al. (1995). No other changes were made to the model or its inputs aside from adding in this one new gas-phase reaction. Results from this scenario were compared to the base case scenario described in the manuscript to determine changes in air quality.

Figure S5 in the SI has been updated to include the results from this new scenario. Additionally, a new figure has been added to the SI (Figure S6, analogous to Figure 11) to show the diurnal profile of indole SOA with and without the inclusion of the indole+$NO_3$ reaction. Additional discussion has been added to the manuscript at the end of the section 3 and in the second to last paragraph in section 4. The changes to the main text are explicitly shown in the revised manuscript using track changes. Figures S5 and S6 are reproduced on the next two pages.

Including the indole+$NO_3$ reaction in the model changes the diurnal profile of both gas-phase indole and indole SOA concentrations. Compared with the high emissions scenario that does not include the indole+$NO_3$ reaction, modeled gas-phase indole concentrations are lower throughout the day (Figure S5). The difference is most pronounced during the early morning and late night hours when $NO_3$ concentrations are highest and OH concentrations are low. There is a corresponding increase in modeled indole SOA concentrations due to increased oxidation of gas-phase indole via reaction with $NO_3$. Modeled indole SOA concentrations are higher throughout the day, although the indole+$NO_3$ reaction contributes to increased indole SOA formation mostly only at night as expected. This causes the relative increase in total SOA concentrations due to indole SOA (Figure S6, right axis) to be more steady throughout the day, rather than peaking during daytime hours when photochemistry is active and OH is available.

**Although the modeled diurnal profiles of indole and indole SOA change when the indole+$NO_3$ reaction is included in the model, the main conclusions of the paper remain unchanged.**

**This is the new Figure S5. Diurnal profile of gas-phase indole concentrations in SoCAB**

[Figure]

**Figure S5: Gas-phase indole concentrations**

The indole mixing ratios in ppbv are shown for the low emissions scenario (a), medium emissions scenario (b), and high emissions scenario (c). In addition, trace (c2) shows mixing ratios in the high emissions scenario with indole oxidation via reaction with $NO_3$ included in the model. Domain wide average concentrations are shown in the top panel and domain maximum concentrations are shown in the bottom panel.

**This is the new Figure S6. Diurnal profile of SOA concentrations in SoCAB**

[Figure]

**Figure S6: Domain wide average SOA concentrations**

The base case mass concentration of SOA is shown in black line referenced to the left axis. Also shown is the percent increase in the domain wide average SOA concentrations (right axis) due to indole SOA in the high emissions scenario (c), and high emissions scenario with indole oxidation via reaction with $NO_3$ included in the model (c2). The daytime concentrations are not significantly affected by $NO_3$ but the nighttime concentrations increase.

[revised manuscript text omitted]